# Directed Graph Grammars for Sequence-based Learning

**Michael Sun** [1]   **Orion Foo** [2]   **Gang Liu** [3]   **Wojciech Matusik** [1]   **Jie Chen** [4]

## Abstract

Directed acyclic graphs (DAGs) are a class of graphs commonly used in practice, with examples that include electronic circuits, Bayesian networks, and neural architectures. While many effective encoders exist for DAGs, it remains challenging to decode them in a principled manner, because the nodes of a DAG can have many different topological orders. In this work, we propose a grammar-based approach to constructing a principled, compact and equivalent sequential representation of a DAG. Specifically, we view a graph as derivations over an unambiguous grammar, where the DAG corresponds to a unique sequence of production rules. Equivalently, the procedure to construct such a description can be viewed as a lossless compression of the data. Such a representation has many uses, including building a generative model for graph generation, learning a latent space for property prediction, and leveraging the sequence representational continuity for Bayesian Optimization over structured data. Code is available at https://github.com/shiningsunnyday/induction.

## 1. Introduction

Directed acyclic graphs (DAGs) underlie many applications in computer science and engineering, from neural architectures (Hutter et al., 2019), Bayesian networks (Koller, 2009), analog circuits, financial transactions, to linearized representations of molecules (Weininger, 1988). Recently, specialized generative models for graphs have been proposed (Li et al., 2018; Simonovsky & Komodakis, 2018; De Cao & Kipf, 2018; Ma et al., 2018; Jin et al., 2018; Liu et al., 2018b; You et al., 2018a; Bojchevski et al., 2018), with well-motivated encoding schemes that respect graph-specific invariances. However, principled solutions for decoding

graphs are still lacking. For example, current methods propose decoding a graph autoregressively by adding nodes and edges, or fragments and connections, at every time step according to some arbitrary ordering (Kusner et al., 2017; Li et al., 2018; Zhang et al., 2019; Thost & Chen, 2021). However, these methods lack rigor and suffer from combinatorial intractability because there can be an exponential number of possible decoding orders. On the other hand, graph grammars, which represent graphs as derivations over a formal language that views subgraphs as "words", have shown enhanced modeling (Jin et al., 2018) and data efficiency (Guo et al., 2022; Sun et al., 2024), but their decoding ability remains limited, lacking behind the progress of generative models for sequential data like natural language. The heart of this issue lies in the absence of a rigorous mapping between the space of graphs and the space of sequences. Given an ideal tokenization strategy, graph modeling reduces to sequence modeling, where innovations like Transformers (Vaswani, 2017) and generative pretraining (Achiam et al., 2023) have made significant progress.

In this work, we propose a novel and faithful mapping that respects several key properties: it is (1) one-to-one and (2) onto over the observed data, (3) deterministic, (4) valid, and (5) strives for Occam's Razor. Our key insight is to parse graphs according to a context-free graph grammar that is constrained to exhibit linear parse trees, producing a sequence of graph rewrite rules that serves as an equivalent, lossless sequential representation for the graph. We implement this mapping for DAGs, using properties specific to DAGs to make the realization of this ideal mapping efficient in practice. Our method, **DI**rected **G**raph **G**rammar **E**mbedded **D**erivations (**DIGGED**) seeks to compress a dataset of given DAGs into parse sequences, incrementally constructing the underlying graph grammar and invoking the principle of Minimum Description Length (MDL). Our contributions include:

- Definitions of the properties for an ideal mapping between DAGs and sequences;
- Novel grammar induction algorithm which respects these properties, with theoretical guarantees;
- Integration within an autoencoder framework for generation, prediction and optimization;
- Comprehensive experiments on real-world applications in neural architectures, Bayesian Networks, and circuits,

---

[1]MIT CSAIL [2]MIT [3]University of Notre Dame [4]MIT-IBM Watson AI Lab, IBM Research. Correspondence to: Michael Sun <msun415@csail.mit.edu>.

*Proceedings of the 42ⁿᵈ International Conference on Machine Learning*, Vancouver, Canada. PMLR 267, 2025. Copyright 2025 by the author(s).

demonstrating better performance and representation quality compared with existing DAG learning frameworks;

- Case studies to interpret and explain our method, highlighting built-in advantages of our method.

By establishing a theoretically motivated mapping between DAGs and sequences, our work offers a new perspective on graph generative modeling and an opportunity to integrate graph data natively into natural language models.

## 2. Related Works

### 2.1. Learning and Optimization of DAGs

DAGs underlie core problems in computer science, such as Bayesian Network structure learning and neural architecture search. Due to the underlying data being discrete (and the problem often NP-hard (Chickering, 1996)), existing works can be categorized into at least one of the following categories: (a) exact search (Singh & Moore, 2005; Yuan et al., 2011; Yuan & Malone, 2013), (b) approximate search (Heckerman et al., 1995; Gao et al., 2017), (c) continuous relaxation (Liu et al., 2018a; Luo et al., 2018; Zheng et al., 2018; Yu et al., 2019), (d) Bayesian Optimization (Yackley & Lane, 2012), and (e) autoencoders (Zhang et al., 2019; Thost & Chen, 2021), with (e) being a modern approach that we adopt. Autoencoders (Kingma, 2013; Rezende et al., 2014) that build a latent space are appealing because they naturally support three downstream tasks: (1) unconditional generation, (2) property prediction from encoded latent embeddings, and (3) optimization over a smooth, continuous space (Zhang et al., 2019). For example, the approach of learning surrogates and optimizing within a smooth continuous space is common in other domains (Mueller et al., 2017; Gómez-Bombarelli et al., 2018). Graph autoencoders that use popular message passing paradigms have gained widespread adoption (Kipf & Welling, 2016; Hamilton et al., 2017), but graph decoders have not evolved beyond strategies that incrementally add atoms/edges or fragments/connections according to an arbitrary order (Li et al., 2018; You et al., 2018b; Zhang et al., 2019; Sun et al., 2024).

### 2.2. Graph Grammars

Graph Grammars (Engelfriet & Rozenberg, 1997; Janssens & Rozenberg, 1982) are precise and formal descriptions of graph transformations. Analogous to string grammars, graph grammars are formal languages that include a vocabulary of subgraphs and a set of rewrite rules. Recently, learning substructure-based graph grammars has become popular for molecules (Jin et al., 2018; Guo et al., 2022; Sun et al., 2024), as motifs like functional groups provide useful abstractions for enhanced interpretability and modeling efficiency over string-based representations (Weininger, 1988).

Despite their usefulness, grammars cannot model every language (Chomsky, 1959), whereas probabilistic Language Models can model arbitrary distributions of sentences. We show how Transformers (Vaswani, 2017) can be adopted for grammar-based graph decoding, combining the best of both worlds.

### 2.3. Concept Induction on Graphs

Concepts, motifs or subgraphs are related ways of expressing patterns on graphs. Unsupervised induction of these patterns takes many forms, but the common theme which guides these approaches is Minimum Description Length (MDL), an example of Occam's Razor. The most common way to achieve the MDL of graphs is Frequent Subgraph Mining (FSM) (Holder, 1989; Gonzalez et al., 2000; Bandyopadhyay et al., 2002). FSM, combined with graph grammar, has practical uses in graph compression (Maneth & Peternek, 2018; Peternek, 2018; Busatto et al., 2004; Peshkin, 2007) and concept discovery (Holder et al., 1994; Holder & Cook, 1993; Cook & Holder, 2000; Djoko et al., 1997; 1995; Cook et al., 1996; 1995; Jonyer et al., 2002), but its use for building modern generative models is unexplored.

## 3. Method

### 3.1. Preliminaries

**Directed Graph Grammar.** Edge-directed Neighborhood Controlled Embedding (edNCE) is a family of formal languages over directed graphs with node labels (and optionally edge labels). Each grammar $G = (\Sigma, N, T, P, S)$ contains a vocabulary of node labels $\Sigma$, vocabulary of edge labels $T$, subset of non-terminal node labels $N \subset \Sigma$, an initial start label $S \in N$, and a set of production rules $P$. A node-labeled directed graph is a tuple $H = (V, E, \lambda)$ where $V$ is the finite set of nodes, $E \subseteq \{(v, \gamma, w) \mid v, w \in V, v \neq w, \gamma \in T\}$ is the set of edges, and $\lambda : V \to \Sigma$ is the node-labeling function. We denote nodes and edges of $H$ as $V_H$ and $E_H$. Each rule is a tuple $(X, D, I)$, with "daughter" graph $D$, applicable to any "host" graph $H$ containing a node $n$, s.t. $\lambda(n) = X \in N$. Applying the rule removes $n$ from $H$, replaces it by a copy of $D$ and embeds it to the remainder of $H$ by forming edges following instructions in $I$. Formally, each instruction is the form $(\sigma, \beta/\gamma, x, d/d')$ $(\sigma \in \Sigma, \beta, \gamma \in T, x \in V_D, d, d' \in \{\text{in}, \text{out}\})$ which when applied to $H$, has the semantics: "establish a $d'$-edge labeled $\gamma$ to node $x$ from each $\beta$-labeled $d$-neighbor with label $\sigma$".

**Terminologies.** Given a dataset $\mathcal{D}$ of node-labeled directed graphs, **induction** is the task of constructing $G$ from data; **parsing** is the task of finding the derivation, for example, the sequence of rules, which produces a given $H$. $G$ is **ambiguous** if there is some data $H_{\text{ambiguous}}$ with two distinct derivations, and $H$ itself is labeled as ambiguous accord-

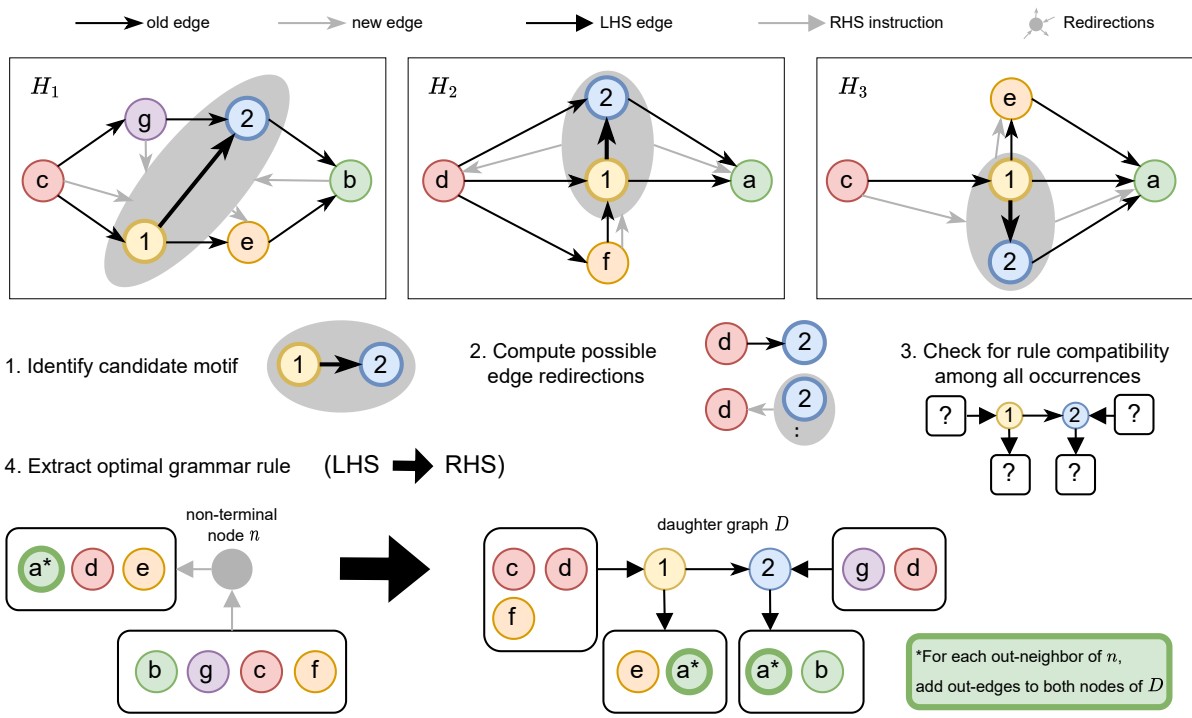

*Figure 1.* We adopt the edNCE grammar formalism. (Top): Dataset $\mathcal{D} = \{H_1, H_2, H_3\}$; (Middle): **Step 1 (Sec 3.2.1).** Our approximate frequent subgraph mining library finds candidate subgraphs. As an example, the induced subgraph from nodes 1 & 2 in all three DAGs is considered. Its occurrences in $H_1, H_2, H_3$ are grounded. **Step 2 (Sec 3.2.2).** For each possible assignment of gray edge directions, bounds on the set of instructions are deduced. For example, the subgraph occurrence in $H_1$ *includes* into $I$, "for each green in-neighbor (gray), add out-edge (black) from node 2", and *excludes* from $I$, "for each green in-neighbor, add out-edge (black) from node 1". $H_2$ includes into $I$: "for each green *out*-neighbor, add out-edges from both nodes 1 and 2". Suppose we had reversed the gray arrow in $H_1$. Then, the exclusion set of case $H_1$ conflicts with the inclusion set of $H_2$, since it's unclear if we should add out-edges from both 1 & 2 to each green *out*-neighbor, or just node 2. Intuitively, cases that differ in the precondition of edge direction are labeled with separate letters (e.g. a vs b), inducing different but non-conflicting instructions. **Step 3 (Sec 3.2.2).** Given bounds on the instruction set for each motif occurrence, the final set of instructions is deduced from the (approximate) solution of a max clique problem. Each node is a (motif occurrence, edge redirections) realization. Each edge indicates compatibility. **Step 4 (Sec 3.2.3).** The candidate motif and the associated solution to Step 3 which minimizes the total data description length is chosen to define a grammar rule. Then, Steps 1-4 are repeated until convergence. (Bottom): A grammar rule consists of a subgraph (gray) and instructions to connect it to the neighborhood. Instructions are grouped by letters, identifying the node label and its directional relationship to the parent gray node.

ing to $G$. $A \Rightarrow B$ represents one rewriting step, and $\overset{*}{\Rightarrow}$ the transitive relation, i.e. **derivation**. The **language** of $G$, denoted as $L(G)$, is the set of non-isomorphic directed graphs $\{H \mid S \overset{*}{\Rightarrow} H\}$. Two directed graphs $H_1, H_2$ are **isomorphic** if there is some bijective mapping $f$ of nodes, $f : V_{H_1} \rightarrow V_{H_2}$ s.t. $(u, v) \in E_{H_1} \Leftrightarrow (f(u), f(v)) \in E_{H_2}$. A subgraph $H'$ of $H = (V, E, \lambda)$ is a tuple $(V', E', \lambda')$ s.t. $V' \subseteq V, E' = \{(v, \gamma, w) \in E \mid v, w \in V'\}, \lambda' : V' \rightarrow \Sigma$ and $\lambda'$ is $\lambda$ restricted to $V'$.

## 3.2. Unsupervised Grammar Induction

Given a dataset $\mathcal{D} = \{H_i \mid i = 1, \dots, |\mathcal{D}|\}$, we create the composite graph $H = (\bigcup_i V_{H_i}, \bigcup_i E_{H_i})$ with $|\mathcal{D}|$ connected components. Through an iterative lossless compression algorithm, the description for $H$ is refined to only $|\mathcal{D}|$

isolated nodes (each with label $S$) and $|\mathcal{D}|$ parses according to its induced grammar $G_{\mathcal{D}}$. We describe the main computation steps of each iteration, emphasizing ideas rather than notation. Further details and pseudocode are in App. B.

### 3.2.1. FREQUENT SUBGRAPH MINING

The first step is to discover common motifs, that is, repetitive instances of the same subgraph, within the current $H$. We adopt the fast, approximate FSM library Subdue (Holder, 1989) on $H$ to obtain a list of common motifs. Our key innovation is to process FSM outputs as follows: for components containing a non-terminal node, *only* subgraphs with that non-terminal node are considered. This simplifies the parse tree to a rooted path. For each motif, we then ground the occurrences by running subgraph isomorphism, parallelized

across connected components of $H$, resulting in a list of occurrences $D_1, D_2, \ldots, D_K$, for each common motif $D$.

### 3.2.2. COMPATIBILITY MAXIMIZATION SOLVER

The second step is to, for each motif, find the maximal subset of occurrences that can be consistent with the same set of connection instructions. In Figure 1, we see each occurrence of the candidate motif includes an incoming edge to node 1 from a red neighbor, so instruction $(c)$ states: "establish an in-edge to node 1 from each in-neighbor with label red". From the second DAG, it appears the same instruction but for node 2 is needed. However, adding such an instruction would create a conflict with the motif's occurrences in DAGs $H_2$ and $H_3$, as the red neighbor doesn't connect to node 2 in those instances. Instead, our compatibility solver finds a different set of instructions (two instructions with group label $(d)$) for which all three occurrences are compatible. Formally, the solver finds the optimal assignment to the variables $\bigcup_{k=1}^{K} \{(d_y, \beta_y) \mid \exists x \in D_k \text{ s.t. } x \text{ neighbors } y\}$ where $d_y \in \{\text{in, out}\}, \beta_y \in T$, representing the direction and edge labels (if any) of the gray edges. The solver is formulated as a maximum-clique problem, where each node represents a possible assignment to $\{(d_y, \beta_y)\}$ for a specific occurrence $k$, and an edge is created between two nodes if the variable assignments they contain are not in conflict with each other. At a high level, each node $v$ carries with it an "inset" and "outset", representing the set of instructions that *must* be present and the set of instructions that *must not* be present, as deduced from the redirection assignments. Determining whether a node exists equates to checking $v_\text{inset} \cap v_\text{outset} = \emptyset$ and whether an edge exists for $u$ and $v$ equates to checking $(u_\text{inset} \cup v_\text{inset}) \cap (u_\text{outset} \cup v_\text{outset}) = \emptyset$ (with some additional minor considerations). After obtaining the clique solution $C := \{v\}$, an or-reduction $\bigcup_{v \in C} v_\text{inset}$ yields the minimal instruction set to *include* in a rule compatible with all occurrences, and similarly $\bigcup_{v \in C} v_\text{outset}$ yields the minimal instruction set to *exclude*. Any instruction set in-between is permissible, and we apply dataset-specific heuristics in selecting the final instruction set for inducing a rule.

### 3.2.3. MINIMUM DESCRIPTION LENGTH

The third step follows after the previous step is repeated over all candidate motifs. We select the solution and its accompanying maximally compatible rule, based on the greedy objective: $\max |C|(|D| - 1)$. The contraction is the reverse operation of a rule application: for each $k$, remove $D_k$, replace it by a non-terminal node $n_k$, and add edges according to the solution's assignment for $\{(d_y, \beta_y)\}$. This step is motivated by prior work that uses MDL as the principle behind unsupervised objectives for graph compression. Assuming the rule has size $O(1)$, the greedy objective is the difference in description length ($\Delta|H|$). The algorithm terminates when $|C| < 2$ over all clique solutions.

### 3.2.4. DISAMBIGUATION PROCEDURE

The final step of grammar induction is to resolve ambiguity in $G$ over $D$ by modifying $G \rightarrow G'$. Preventing this in general is impossible because determining whether a given graph grammar $G$ is ambiguous is undecidable (see App. C). Nevertheless, we can find all derivations for a given graph $H_i \in \mathcal{D}$. This problem, in general, is NP-hard (Engelfriet & Rozenberg, 1997). We present a dynamic parsing programming algorithm that is the DAG counterpart to the well-known CYK algorithm (Cocke, 1969; Younger, 1967; Kasami, 1966) and takes advantage of two properties specific to DAGs and our grammar. The first exploits the theoretical insight that DAGs have canonical string representations (more in App. D), enabling hashing-based memoization. The second prunes intermediate graphs that become disconnected or cyclic, as those are not valid intermediate results (Deterministic property). After finding all derivations, we find the minimal set of rules that, when removed from $G$, leaves the largest subset of $\mathcal{D}$ with one unique derivation over $G'$. The formulation is in terms of the maximum hitting set problem. The algorithmic details are in App. C.

### 3.3. Properties

We elaborate on how DIGGED addresses the limitations of existing methods (Table 1). We will analyze two broad categories of methods: autoregressive generation (AG), which builds up a graph incrementally, tracking the intermediate graph to decide the next action, and sequential decoding (SD), which directly generate descriptors that encode the adjacency information of the graph using some permutation of the nodes.

*Table 1.* DIGGED offers comprehensive guarantees that existing methods fail to or partially address.

| Methods | One-to-one? | Onto? | Deterministic? | Valid? | Stateless? |
|---------|-------------|-------|----------------|--------|------------|
| AG | ✗ | ✓ | ✗ | ✓ | ✗ |
| SD | ✗ | ✓ | ✓ | ✗ | ✓ |
| **DIGGED** | ✓ | ✓ | ✓ | ✓ | ✓ |

1. **One-to-one (over $\mathcal{D}$).** For every $H \in \mathcal{D}$, our unambiguous procedure assures there is only one way to parse it. AG methods can generate the same graph in many (up to exponential) ways. SD methods rely on an arbitrary ordering of the nodes.
2. **Onto (over $\mathcal{D}$).** The proof in the appendix shows that the grammar induction algorithm is a concurrent parsing algorithm for each $H \in \mathcal{D}$, so $\mathcal{D} \subseteq L(G)$. Both AG and SD can generate any graph.
3. **Deterministic.** That reconstruction is deterministic follows immediately from properties of the grammar. We also show additional desiderata, namely that each inter-

mediate graph is always an *unambiguous DAG*, respecting the properties of $L(G)$. AG methods can take many action trajectories to arrive at the same final state.

4. **Validity.** edNCE grammars are context-free, so an arbitrary derivation still produces a valid directed graph. An arbitrary adjacency string (SD) is not guaranteed to encode a valid graph.

5. **Stateless.** Context-free grammars are stateless. Generation terminates when a selected rule contains no further non-terminals. AG methods require tracking the intermediate graph as the state, to filter out invalid actions.

See App. A for full proofs of above properties 1-4 and more remarks. DIGGED also meets two soft desiderata that are appealing to downstream use cases.

- **Controllable**. Due to a context-free, sequential representation, it is easy to add context-sensitive constraints at each step to enforce domain-specific validity. We use a real-world example in Section 3.4.
- **Compositional**. Each DAG is a compact program composed from reusable primitives learned for efficient lossless compression. Compositionality provides a lens to understand generalization on downstream tasks, as elaborated on in App. G.

### 3.4. Sequence-based Learning

Once we have converted each $H \in \mathcal{D}$ into a sequential description, we jointly train an encoder and decoder within an autoencoder framework. As visualized in Figure 2, we decode a sequence of individual rules, which, when concatenated together, reconstructs the input. Standard to VAE training, we maximize the evidence lower bound. See App. I for hyperparameters used.

**Graph Encoder (Option 1).** We support using DAGNN (Thost & Chen, 2021) as an encoder from DAG to latent space. This combines existing SOTA architectures for DAG-specific encoding, while supporting the additional use case of parsing DAGs $\notin L(G)$ to a similar DAG $\in L(G)$.

**Rule Sequence Encoder (Option 2).** We also support a Transformer encoder with full attention to encode a sequence of rule tokens to latent space. Simultaneously, we learn a dictionary of embeddings, one for each rule, as is standard for generative pretraining (Achiam et al., 2023).

We include comparisons between these two options in our experiments, where we show a GNN encoder constructs a richer latent space for unconditional generation and property prediction. Therefore, we use a GNN encoder for obtaining the final results and analyses. Analogous to progress in language modeling, we believe a full attention Transformer encoder is the natural and scalable approach. For instance, we show rule token frequency also follows Zipf's Law (Zipf, 2013). Please refer to App. H for an illuminating discussion.

**Rule Sequence Decoder.** We adopt a Transformer decoder with causal attention masking to autoregressively decode a sequence of rule tokens from latent space. Due to the one-to-one guarantee, reconstruction equates to exact match of the sequence. During training, we pad each rule sequence to the maximum length in the batch and do batched cross-entropy loss. We jointly train the encoder and decoder using standard reconstruction and KL divergence loss.

**Inference.** Our edNCE grammar is context-free (Engelfriet & Rozenberg, 1997), so each rule can be independently applied one after another. On the first step, we mask out all rules whose LHS is not $S$. On subsequent steps, we mask out all rules whose LHS label is not the same as the current non-terminal node. The sampling terminates when there are no more non-terminals. Facilitated by deterministic decoding, we can constrain the sampling to guarantee validity. For example, in analog circuits, we can ensure only valid op-amps are decoded, because the stabilization constraint (each +gm- and -gm- transconductance unit must be in parallel with a resistor and capacitor) can be translated into a predicate over the set of *new* nodes and edges that would be introduced by each rule. These incremental validity checks ensure inference remains efficient, while showcasing our context-free grammar's flexibility for domain-specific customization (see Sec. 6.4 for a case study).

## 4. Experiments

### 4.1. Datasets

1. **Neural Architectures (ENAS).** The ENAS dataset contains 19,020 neural architectures from the ENAS software and their weight-sharing accuracy (WS-Acc) on CIFAR10 (Pham et al., 2018). We compare with the same baselines reported in Thost & Chen (2021).

2. **Bayesian Networks (BN).** The BN dataset contains 200,000 random, 8-node Bayesian networks from the R package bnlearn (Scutari, 2009) and their Bayesian Information Criterion (BIC) score for fitting the Asia dataset (Lauritzen & Spiegelhalter, 1988). We compare with the same baselines as for ENAS.

3. **Analog Circuits (CKT).** The CKT dataset contains 10000 operational amplifiers (op-amps) released by Dong et al. (2023) and their simulated metrics: gain, bandwidth (BW), phase margin (PM), and figure of merit (FoM). We compare with the same baselines reported in Dong et al. (2023).

### 4.2. Task Setup

1. **Unconditional Generation.** For unconditional generation, we sample from a Gaussian prior. For each latent point, we perform constrained decoding of a sequence of rules, then derive the DAG. For all datasets, we evaluate

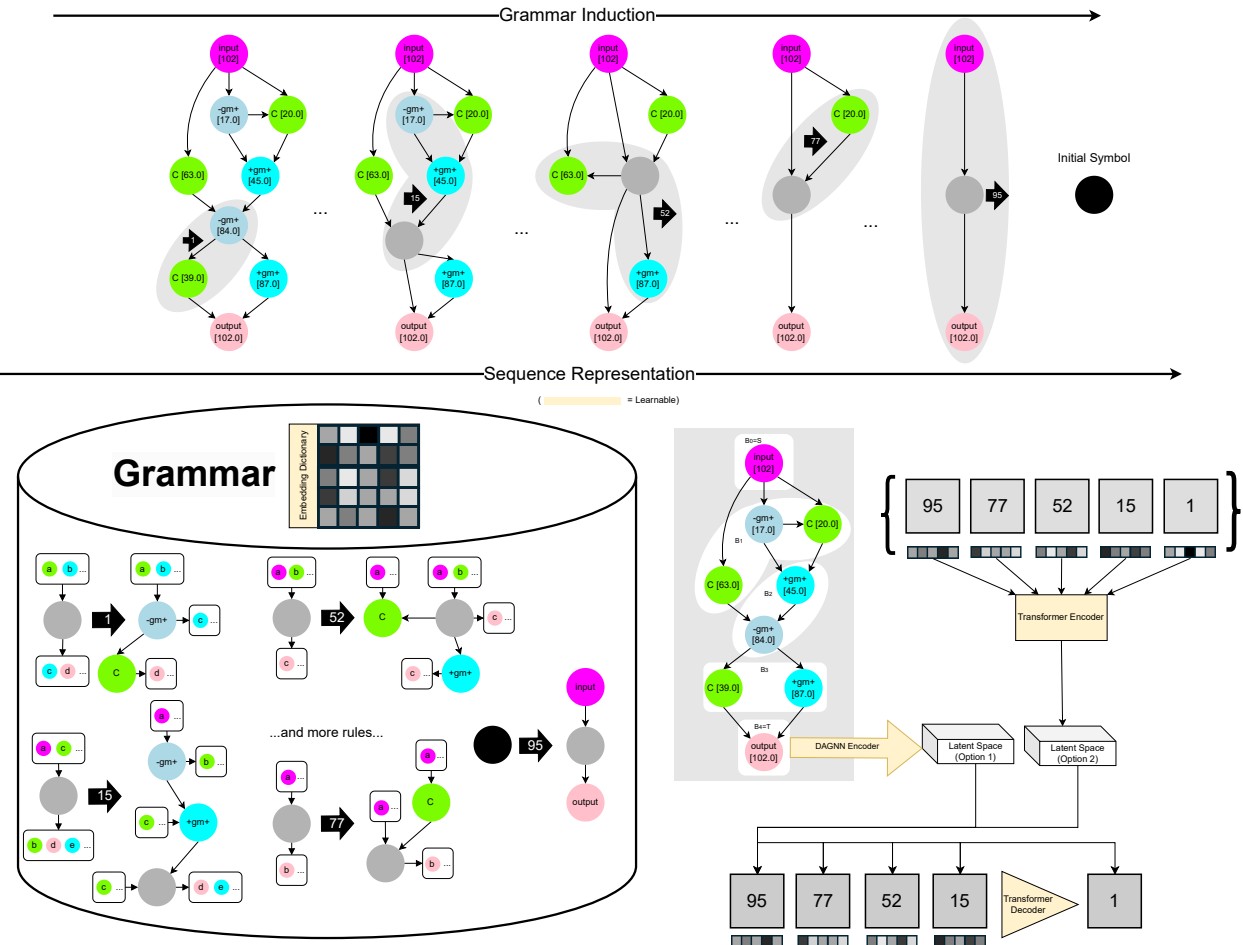

*Figure 2.* (Top) Our grammar induction framework iteratively minimizes the total description length of $\mathcal{D}$, contracting common and compatible motifs, producing grammar rules while parsing the input according to the grammar. (Bottom-left) Our induction algorithm builds the token dictionary, where individual rules are the tokens used in a faithful sequential representation of the DAG. (Bottom-right) We experiment with two ways to encode the DAG: 1) using a full attention Transformer encoder vs 2) using a GNN tailored to DAGs (Thost & Chen, 2021); in both cases, we use causal, autoregressive Transformer decoder within an autoencoder framework, while jointly learning the embedding dictionary.

the reconstruction, validity and novelty. For circuits, we also evaluate circuit validity, defined in the same way as Dong et al. (2023).

2. **Predictive Performance.** For property prediction, we train a Sparse Gaussian Process (SGP) regressor, following the same setup and hyperparameters as Zhang et al. (2019); Thost & Chen (2021); Dong et al. (2023).

3. **Bayesian Optimization.** We run batched Bayesian Optimization based on the SGP model for 10 rounds with 50 acquisition samples per round. We follow the same setup as Zhang et al. (2019) for ENAS and BN and Dong et al. (2023) for CKT, reproducing the same Cadence SPECTRE simulation environment, adopting the same DAG-to-netlist conversion logic, and run the same simulation script.

### 4.3. Baselines.

We compare with prior AG methods and SD methods (see Sec. 3.3 for descriptions). Methods under both categories can be analyzed by how nodes are ordered: S-VAE, D-VAE and DAGNN use topological order; PACE uses canonical order; GraphRNN uses BFS order; CktGNN defines a total order on a basis of subcircuits. Transformer-based methods (Graphormer and PACE) rely on a well-chosen ordering and use positional encoding to improve encoding efficiency. These baselines choose various ordering criteria to meet the one-to-one property, via a traversal algorithm or canonicalization. We hypothesize these steps do not overcome inherent limitations in the representation. We show DIGGED's theoretically sound and compact sequential descriptions translate into practical performance advantages.

# 5. Results

*Table 2.* Prior validity, uniqueness and novelty (%). We follow the same settings as Zhang et al. (2019).

| Methods | Neural architectures | | | | Bayesian networks | | | |
|---|---|---|---|---|---|---|---|---|
| | Accuracy | Validity | Uniqueness | Novelty | Accuracy | Validity | Uniqueness | Novelty |
| D-VAE | 99.96 | **100.00** | 37.26 | **100.00** | 99.94 | 98.84 | 38.98 | 98.01 |
| S-VAE | 99.98 | **100.00** | 37.03 | 99.99 | 99.99 | **100.00** | 35.51 | 99.70 |
| GraphRNN | 99.85 | 99.84 | 29.77 | **100.00** | 96.71 | **100.00** | 27.30 | 98.57 |
| GCN | 98.70 | 99.53 | 34.00 | **100.00** | 99.81 | 99.02 | 32.84 | 99.40 |
| DeepGMG | 94.98 | 98.66 | 46.37 | 99.93 | 47.74 | 98.86 | 57.27 | 98.49 |
| DIGGED (GNN) | **100** | **100** | **98.7** | 99.9 | **100** | **100** | 97.6 | **100** |
| DIGGED (TOKEN) | **100** | **100** | 25.4 | 37.8 | **100** | **100** | **98.67** | 26.67 |

*Table 3.* Effectiveness in real-world electronic circuit design. Training data is CktBench101 (Dong et al., 2023) for all baselines except top group. CktGNN also has an option to use CktBench301 as pivots in the BO. We also include top 90/95/max designs from CktBench101 and CktBench301.

| Methods | Valid DAGs (%) ↑ | Valid circuits (%) ↑ | Novel circuits (%) ↑ | BO (FoM) ↑ |
|---|---|---|---|---|
| PACE | 83.12 | 75.52 | 97.14 | 33.2742 |
| DAGNN | 83.10 | 74.21 | **97.19** | 33.2742 |
| D-VAE | 82.12 | 73.93 | 97.15 | 32.3778 |
| GCN | 81.02 | 72.03 | 97.01 | 31.6244 |
| GIN | 80.92 | 73.17 | 96.88 | 31.6244 |
| NGNN | 82.17 | 73.22 | 95.29 | 32.2827 |
| Graphormer | 82.81 | 72.70 | 94.80 | 32.2827 |
| CktGNN | 98.92 | 98.92 | 92.29 | 33.4364 |
| CktGNN (CktBench301) | — | — | — | 190.2354 |
| CktBench101 (90%, 95%, max) | 100 | 100 | 0 | 186.3870 233.1829 326.6657 |
| CktBench301 (90%, 95%, max) | 100 | 100 | 100 | 90.8379 119.9001 197.2296 |
| DIGGED (GNN) | **100** | **100** | 78.80 | **310.2635** |
| DIGGED (TOKEN) | 92.2 | 92.2 | 60.7 | — |

## 5.1. Unconditional Generation

**ENAS & BN.** Shown in Table 2, DIGGED ensures Validity and achieves near 100% Uniqueness on ENAS and BN, > 50% and > 40% higher than the second best method. On BNs, it's the only method achieving 100% Novelty, showing ability to sample diverse, combinatorial structures.

**CKT.** Shown in Table 3, DIGGED ensures 100% Validity both at the syntax (DAG) and semantic (circuit) level, serving as a powerful complement to synthetic data generation pipelines.

## 5.2. Predictive Performance

**CKT.** As shown in Table 4, DIGGED produces discriminative latent representations when combining a dedicated DAG encoder with sequence-based decoding with Transformers. It achieves 26.5% lower RMSE and 60% higher Pearson $r$ on the holistic metric, FoM, over the next best (CktGNN), which is a domain-specific GNN that uses hand-selected motifs to form a subgraph basis.

**ENAS.** As shown in Table 5, DIGGED slightly underperforms the best generative model encoder (DAGNN). We suspect that this is due to the large number of rules (7504) in grammar, making dictionary learning cumbersome.

**BN.** We observe an interesting case of high Pearson $r$ but a more modest RMSE. We conduct a closer, visual, and quantitative investigation of this result in App. F, showing a global, linear trend. We believe this to be a consequence of our sequence learning framework, where there is representation continuity in the latent space. This showcases the downstream representation learning advantages of training a modern Transformer architecture on a principled and congruous sequence representation.

## 5.3. Bayesian Optimization

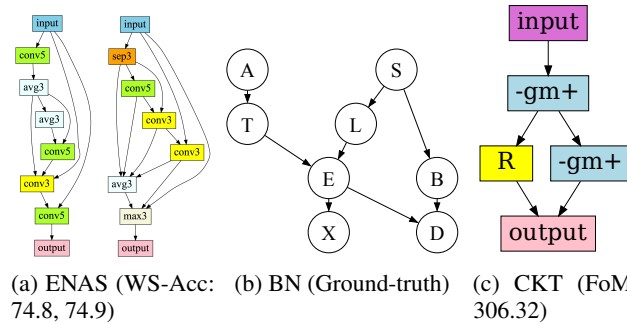

(a) ENAS (WS-Acc: 74.8, 74.9)  (b) BN (Ground-truth)  (c) CKT (FoM: 306.32)

*Figure 3.* We visualize the best discovered designs from BO. We reproduce the same BO and evaluation setup as Zhang et al. (2019); Pham et al. (2018); Dong et al. (2023).

In Figure 3, we visualize the best, *novel* designs found during BO.

**CKT.** DIGGED generated *novel* designs that exceeded the best design in CktBench301, with the best one only 5% lower in FoM than the best design in CktBench101. Visualized in Fig. 3, we see a simple but effective double-stage amplifier, with a parallel resistor configuration, with a FoM of 306.32. We visualize additional designs in App. E, and observe that they all have short derivation lengths, implicitly capturing *simplicity*, an essential requirement for real-world circuit design. More details on baselines are in App. E.3.

**ENAS.** In Fig. 3, we see a novel architecture that combines the overall topology of the best designs found by Zhang et al. (2019) with the consecutive avg. pooling layer design found by Bowman et al. (2015). We also recover one of the best (top 1%) designs in the dataset, with a weight-sharing accuracy of 74.9. This shows the model is versatile, able to reconstruct existing designs and combine aspects of designs found by different previous models.

**BN.** In Fig. 3, we were able to recover *all* the dependencies in the ground-truth model ((Lauritzen & Spiegelhalter, 1988)). This is impressive considering that DIGGED discovered it on the 5th round of BO.

*Table 4.* Predictive Performance of Latent Representations on CktBench101. We follow the same settings as Dong et al. (2023).

| Evaluation Metric | Gain | | BW | | PM | | FoM | |
|---|---|---|---|---|---|---|---|---|
| | RMSE ↓ | Pearson's r ↑ | RMSE ↓ | Pearson's r ↑ | RMSE ↓ | Pearson's r ↑ | RMSE ↓ | Pearson's r ↑ |
| PACE | 0.644 ± 0.003 | 0.762 ± 0.002 | 0.896 ± 0.003 | 0.442 ± 0.001 | 0.970 ± 0.003 | 0.226 ± 0.001 | 0.889 ± 0.003 | 0.423 ± 0.001 |
| DAGNN | 0.695 ± 0.002 | 0.707 ± 0.001 | 0.881 ± 0.002 | 0.453 ± 0.001 | 0.969 ± 0.003 | 0.231 ± 0.002 | 0.877 ± 0.003 | 0.442 ± 0.001 |
| D-VAE | 0.681 ± 0.003 | 0.739 ± 0.001 | 0.914 ± 0.002 | 0.394 ± 0.001 | **0.956 ± 0.003** | 0.301 ± 0.002 | 0.897 ± 0.003 | 0.374 ± 0.001 |
| GCN | 0.976 ± 0.003 | 0.140 ± 0.002 | 0.970 ± 0.003 | 0.236 ± 0.001 | 0.993 ± 0.002 | 0.171 ± 0.001 | 0.974 ± 0.003 | 0.217 ± 0.001 |
| GIN | 0.890 ± 0.003 | 0.352 ± 0.001 | 0.926 ± 0.003 | 0.251 ± 0.001 | 0.985 ± 0.004 | 0.187 ± 0.002 | 0.910 ± 0.003 | 0.284 ± 0.001 |
| NGNN | 0.882 ± 0.004 | 0.433 ± 0.001 | 0.933 ± 0.003 | 0.247 ± 0.001 | 0.984 ± 0.004 | 0.196 ± 0.002 | 0.926 ± 0.002 | 0.267 ± 0.001 |
| Pathformer | 0.816 ± 0.003 | 0.529 ± 0.001 | 0.895 ± 0.003 | 0.410 ± 0.001 | 0.967 ± 0.002 | 0.297 ± 0.001 | 0.887 ± 0.002 | 0.391 ± 0.001 |
| CktGNN | **0.607 ± 0.003** | **0.791 ± 0.001** | 0.873 ± 0.003 | 0.479 ± 0.001 | 0.973 ± 0.002 | 0.217 ± 0.001 | 0.854 ± 0.003 | 0.491 ± 0.002 |
| DIGGED (GNN) | 0.630 ± 0.005 | 0.771 ± 0.004 | **0.635 ± 0.006** | **0.784 ± 0.001** | 0.990 ± 0.001 | **0.314 ± 0.001** | **0.627 ± 0.002** | **0.787 ± 0.001** |
| DIGGED (TOKEN) | — | — | — | — | — | — | 1.005 ± 0.0002 | 0.199 ± 0.001 |

*Table 5.* Predictive performance of latent representation on ENAS & BN test set. We follow same settings as Zhang et al. (2019).

| Model | ENAS | | BN | |
|---|---|---|---|---|
| | RMSE ↓ | Pearson's r ↑ | RMSE ↓ | Pearson's r ↑ |
| S-VAE | **0.644 ± 0.003** | **0.762 ± 0.002** | 0.896 ± 0.003 | 0.442 ± 0.001 |
| GraphRNN | 0.695 ± 0.002 | 0.707 ± 0.001 | **0.881 ± 0.002** | 0.453 ± 0.001 |
| GCN | 0.681 ± 0.003 | 0.739 ± 0.001 | 0.914 ± 0.002 | 0.394 ± 0.001 |
| DeepGMG | 0.976 ± 0.003 | 0.140 ± 0.002 | 0.970 ± 0.003 | 0.236 ± 0.001 |
| D-VAE | 0.890 ± 0.003 | 0.352 ± 0.001 | 0.926 ± 0.003 | 0.251 ± 0.001 |
| DAGNN | 0.882 ± 0.004 | 0.433 ± 0.001 | 0.933 ± 0.003 | 0.247 ± 0.001 |
| DIGGED (GNN) | 0.912 ± 0.001 | 0.386 ± 0.001 | 0.953 ± 0.052 | **0.712 ± 0.013** |
| DIGGED (TOKEN) | 0.987 ± 0.001 | 0.049 ± 0.006 | 0.989 ± 0.0001 | 0.129 ± 0.002 |

*Table 6.* Results of our controlled study comparing with simpler node-order encodings. Only FoM is reported for CKT.

| | | Valid | Unique | Novel | RMSE | Pearson's r | 1st | 2nd | 3rd |
|---|---|---|---|---|---|---|---|---|---|
| Graph2NS-Default | ENAS | 96.1 | 99.17 | 100 | 0.746 | 0.656 | 0.746 | 0.744 | 0.743 |
| | BN | 95.8 | 96.4 | 94.8 | 0.498 | 0.869 | -11590 | -11685 | -11991 |
| | CKT | 80.2 | 71.0 | 96.8 | 0.695 | 0.738 | 220.96 | 177.29 | 148.92 |
| Graph2NS-BFS | ENAS | 40.8 | 100 | 100 | 0.806 | 0.595 | 0.746 | 0.746 | 0.745 |
| | BN | 2.2 | 100 | 100 | 0.591 | 0.819 | -11601 | -11892 | -11950 |
| | CKT | 0.1% | 100 | 100 | 0.676 | 0.751 | - | - | - |
| Graph2NS-Random | ENAS | 0% | - | - | 0.859 | 0.508 | - | - | - |
| | BN | 8.4 | 100 | 100 | 0.535 | 0.857 | -11523 | -11624 | -11909 |
| | CKT | 0% | - | - | 0.680 | 0.760 | - | - | - |
| DIGGED | ENAS | 100 | 98.7 | 99.9 | 0.912 | 0.386 | 0.749 | 0.748 | 0.748 |
| | BN | 100 | 97.6 | 100 | 0.953 | 0.712 | -11110 | -11250 | -11293 |
| | CKT | 100 | 100 | 78.8 | 0.627 | 0.787 | 306.32 | 296.82 | 265.53 |

# 6. Discussion

## 6.1. Ablation Study on Simpler Sequential Descriptions.

We perform a controlled ablation in Table 6 fixing DAGNN as the encoder and the same Transformer decoder architecture used to train DIGGED. We vary various node-order encodings as the output targets to test whether simpler SD encodings suffice. We target three node-orderings – default order (that is, the order provided by the data, normally a topological order with domain-specific criteria), BFS from a randomly chosen seed node (as in You et al. (2018b)), or a random order – for comparison. We see the default order is unique in most cases, but its unguaranteed validity results in lower BO optimization results (following Zhang et al. (2019) to deal with invalid samples). BFS or random ordering destroys the decoder's ability to generate valid examples. BFS is do-able for mostly linear path graphs in ENAS but is entirely infeasible for BNs, due to dense dependencies making the order unpredictable. Imposing position on inherently position-invariant graphs causes decoding failures – even DAGs can admit exponentially many valid orders. DIGGED instead is position-*less*; it learns a unique, position-free sequential "change-of-basis" that encodes a graph as its construction steps. Each token includes a set of instructions to recreate the graph. For further explanations and deeper analysis, please refer to App. G.

## 6.2. Ablation Study on Speed-Accuracy Elasticity.

For each solver module for the sub-problems in Sec. 3, we offer brute force, approximation, and heuristic algorithms.

Since subgraph isomorphism, max clique, and hitting set are all NP-hard, we toggle between brute, approximate or heuristic solvers based on problem size. We state our choice of approximation and heuristic variants, along with any hyperparameters to control solution quality, in App. B. We anticipate the setting of larger data sizes, where faster solvers must be chosen out of necessity. Our main results on CKT (the smallest dataset) already reflect exact solutions or high-quality approximations, so we use this dataset to benchmark the performance & efficiency impact of using coarser approximations. We separately conduct four possible changes to the DIGGED algorithm: (1) For Subdue (FSM), we use beam_width = 3 instead of 4; (2) for max clique, we use the greedy algorithm with $K = 10$ random starting nodes; (3) for hitting set, we do beam search with beam_width = 10 instead of exact; (4) we skip disambiguation for early convergence. We find Ablation 3 did not introduce meaningful changes, as beam search equates to an exact procedure for small input sizes. Table 7 shows Ablations 1, 2, 4 speed up execution with minimal quality loss; max clique offers the best accuracy/speed trade-off. Latent space quality and top 3 results benefit slightly from more accurate FSM and max clique solutions, but results are still reasonably close.

## 6.3. Case Study on Lossless Compression Rate.

We empirically analyze how much the total size $|H|$ is compressed relative to the number of rules. We see in Fig. 4 that DIGGED achieves 2.2%, 2.6%, 1.56% compression ratio (initial $|H|$ to pre-termination $|H|$, when every initial

*Table 7.* Results of ablation study, quantifying speed-accuracy tradeoffs for each module. RMSE ↓ (left) and Pearson $r$ ↑ (right) is reported for FoM. Compress ratio is defined in Sec. 6.3.

|         | Unique | Novel | FoM |       | 1st    | 2nd    | 3rd    | %Faster | Compress Ratio |
|---------|--------|-------|-------|-------|--------|--------|--------|---------|----------------|
| Abl.1   | 65.6   | 69.1  | 0.624 | 0.786 | 267.55 | 253.61 | 246.78 | 562%    | 2.04           |
| Abl.2   | 91.3   | 85.1  | 0.617 | 0.797 | 278.93 | 278.93 | 267.61 | 1844%   | 2.13           |
| Abl.3   | 97.3   | 100   | 0.625 | 0.785 | 306.32 | 290.42 | 260.97 | ∼300%   | 2.32           |
| DIGGED  | 98.7   | 99.9  | 0.627 | 0.787 | 306.32 | 296.82 | 265.53 | 0%      | 2.18           |

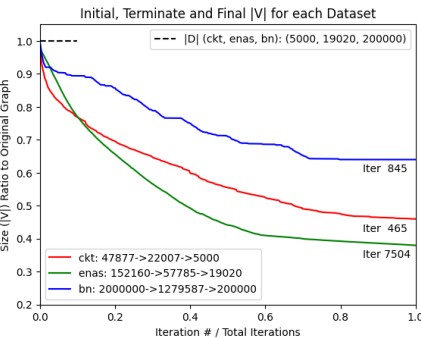

*Figure 4.* We show $M := |H|$ as a function of iteration (same as the number of rules induced). Axes are scaled to 1.0 for standardization across datasets. The lower legend follows the format initial $|H| \rightarrow$ pre-termination $|H| \rightarrow$ post-termination $|H|$ (=$|\mathcal{D}|$).

connected component is contracted to a single node). For convenience, we only show compression for the initial call to Algo. 2 (iter = 0 in Algo. 1), prior to disambiguation. We observe the trend: *linear structures tend to achieve greater total compression ratio at the tradeoff of higher grammar complexity*. For example, ENAS DAGs are linear path-like graphs (with a few skip connections), whereas BN DAGs are graphical models with highly interconnected topologies. CKT DAGs are somewhere in between, with main stages lined up consecutively but also intricate, parallel configurations. Thus, we see compression ratio from highest to lowest: ENAS, CKT, BN. For BN, we see a small (845) number of rules relative to its total size (200k) responsible for a large compression ratio. Intuitively, DIGGED uses the neighborhood topology to deduce a maximally compatible instruction set, so simpler neighborhood topologies like those found in ENAS graphs makes achieving compatibility across occurrences easier, resulting in much more rules (7504). Meanwhile, complex neighborhood topologies in BN may be inherently incompatible with any rule, so there exists some limit on how much compression is possible.

### 6.4. Case study on Real-World Use Case.

DIGGED successfully generates high-performing analog circuit designs by inducing a data-driven grammar, balancing generalization with domain specificity. The case study (App. E) shows its ability to optimize op-amp topologies, where traditional methods focus on device sizing for fixed

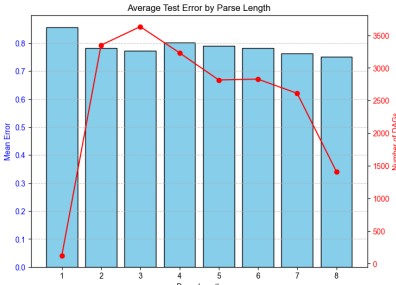

*Figure 5.* We stratify the test error distribution across the parse length. For reference, we also include a count of the number of test set examples of each parse length.

topologies, and existing graph-based approaches rely on predefined substructures. DIGGED constructs designs step-by-step, enforcing meaningful constraints that ensure stability and explainability (App. E.1). Expert evaluation of the highest-performing circuits confirms the validity of many designs while highlighting areas for refinement (App. E.2). Compared to standard black-box optimization baselines, DIGGED's grammar-guided search provides interpretable solutions with improved structural integrity (App. E.3).

### 6.5. Case study on Representation Continuity.

We also use BN as a case study for the relationship between per-sample compression, i.e. the length of its rule sequence, and downstream predictive performance, i.e. the error from a fitted SGP regressor. In Fig. 5, we find that longer rule sequences are more informative, resulting in an inverse relationship between the description length and the test error. By viewing grammar induction as lossless compression (Section 6), we can use the length as a rough estimate of per-sample compression ratio. The BN dataset is also apt for this study because every DAG has a fixed set of nodes, so we don't need to normalize for the initial size $|H|$. We find in Fig. 5 that the representations, in general, become more discriminative with longer parses. We believe this is attributed to compression being an explicit form of information bottleneck (Tishby et al., 2000), where our MDL-guided compression explicitly optimizes for representation compactness, via compositionality, to form a richer representation space amenable for downstream tasks.

## 7. Conclusion

We introduce DIGGED, a principled and efficient mapping from DAGs to sequences via graph grammar parsing. The resulting compact, unambiguous derivations enable a one-to-one problem mapping to sequence modeling. Experiments on real-world optimization problems demonstrate superior performance. An exciting direction is to explore compositional reasoning capabilities with DIGGED representations.

## Acknowledgements

Michael and Gang completed internships at the MIT-IBM Watson AI Lab. Gang is supported by the IBM Fellowship. The authors thank Dr. Xin Zhang from IBM Research for helpful discussions on circuit design use case.

## Impact Statement

This paper presents work whose goal is to advance the field of Machine Learning. There are many potential societal consequences of our work, none which we feel must be specifically highlighted here.

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

# A. Grammar Properties

In this section, we discuss, at a high-level, the nice properties of our grammar-induced sequence representation for DAGs.

**One-to-one.** Theorem C.2 shows that testing if a grammar is one-to-one is, in general, undecidable. Instead, we resort to using the data itself as "test cases" for ambiguity. If there exists two derivations for some $H \in \mathcal{D}$, we can use Algorithm 4 to disambiguate the grammar by removing a minimal set of rules that leaves $H$ unambiguous.

**Proof.** in App. C.

**Onto.** Our mapping is onto $\mathcal{D}$ by construction because our unsupervised grammar induction algorithm simultaneously outputs a parse of each $H \in \mathcal{D}$. This parse is equivalent to lossless compressed representation of $\mathcal{D}$. Details are in App. B.

**Deterministic.** In addition to ensuring $H \in \mathcal{D}$ being onto and one-to-one w.r.t. to its parse, grammar also, by definition ensures that each intermediate graph can be reconstructed. i.e. $H'$ s.t. $S \overset{*}{\Rightarrow} H' \overset{*}{\Rightarrow} H \in \mathcal{D}$ is also one-to-one and onto over $\mathcal{D}$. Furthermore, we show the following.

**Lemma.** It is possible to restrict each intermediate graph, i.e. $H'$ s.t. $S \overset{*}{\Rightarrow} H' \overset{*}{\Rightarrow} H$ to be unambiguous.

**Proof.** I claim that given a parse for $H \in \mathcal{D}$ of the form $S \ldots \overset{p_t}{\Rightarrow} H^{(t)} \ldots \Rightarrow H$, $H^{(t)}$. If $H^{(t)}$ was ambiguous, then clearly $H$ is ambiguous too because the first $t - 1$ steps can be replaced with a different parse. This is a contradiction to the one-to-one property.

**Lemma.** It is possible to restrict each intermediate graph, i.e. $H'$ s.t. $S \overset{*}{\Rightarrow} H' \overset{*}{\Rightarrow} H$ to be a DAG during grammar induction.

**Proof.** We proceed by induction.

Denote $H$ as $H^{(T)}$, where $T$ is the number of steps in the derivation. $H^{(T)} \in \mathcal{D}$ is by assumption a DAG.

Suppose $H^{(t)}$ is a DAG during Algorithm 1. Then, we show $H^{(t-1)}$ can also be constrained to be a DAG, based on the redirections chosen. Let $S$ be the subgraph we contract. Since $S$ is a subgraph, it is also a DAG. There can also be no cycles in $H^{(t)}(V_{H^{(t)}} \setminus V_S)$. Then, we can choose choose every redirection to be "out" or every redirection to be "in" relative to the neighborhood of $S$. This way, a cycle cannot form using both nodes in $S$ and in $V_H^{(t)}$.

**Remarks.** In practice, we don't restrict every redirection to be the same (either "out" or "in"). We compute a "precedence graph" using the nodes in the neighborhood of $S$, based on reachability *without* $S$, i.e. a path finding algorithm with $S$ as the obstacles. Then, we ensure the redirections don't violate this precedence graph. Intuitively, the precedence graph itself is a DAG (otherwise there's a cycle), so there are many permissable redirection sets.

# B. Grammar Induction Algorithm

We delve deeper into the key computation steps of Algorithm 2. The first subsection discusses our implementation choices for the approximate and heuristic variants of the frequent subgraph mining ("fast_subgraph_isomorphism"), max clique ("approx_max_clique"), and hitting set ("quick_hitting_set") problems. The second and third subsections elaborate further on the insets_and_outsets and find_iso functions. They are non-standard problems that we formulated so we feel they require further elaboration.

## B.1. Solver Options

1. Frequent subgraph mining

   (a) Approximate: We use the Subdue library. It has various options for pruning the search. Parameter: beam_width (used for subgraph expansion).

2. Max clique

   (a) Exact (O(exp(n))): networkx's cliques library
   (b) Approximate (O(poly(n))): We use networkx's $O(|V|/(\log |V|)^2)$ approximation algorithm.
   (c) Heuristic (O(n)): (Repeat K times) Initialize a random node, iterate over all remaining nodes in random order, adding any that satisfies clique condition. Parameters: K

3. Hitting set problem during disambiguation

(a) Exact: our own implementation

(b) Approximate: Beam search. Parameters: beam_width

Our datasets have variable sizes from 47877 (CKT), 152160 (ENAS), to 2,000,000 nodes (BN), which span the range of real-world use cases. We use the size of the input to toggle between different options, trading off accuracy and efficiency. Roughly speaking, CKT mostly uses exact/approximate solutions, ENAS approximate/heuristic solutions and BN heuristic solutions.

## B.2. Compute Insets and Outsets

To understand why Algorithm 1 losslessly compresses $\mathcal{D}$, we must understand what the function insets_and_outsets does in the logic of Algorithm 3. The concept of insets and outsets were introduced in (Blockeel & Nijssen, 2008) for a simpler grammar formalism, but we extend it to general edNCE grammars. Here, we restate it here for completeness. Given a subgraph $S$ in a graph $G$, we need to infer $I$, the set of instructions that can induce a grammar rule while ensuring $G$ can be reconstructed. Recall that each instruction in $I$ is of the form $(\sigma, \beta/\gamma, x, d/d')$ which has the semantics "if a neighbor has edge direction $d$, edge label $\beta$, and label $\sigma$, form an edge with direction $d'$ labeled $\gamma$ to node $x \in V_S$" during a one-step derivation. Thus, each instruction carries a precondition $(d, \beta, \sigma)$ and postcondition $(d', \gamma, x)$. Given $G$, $S$, and a *possible* realization $G'$ ($G$ but with $S$ replaced with a non-terminal), we can immediately deduce which (precondition, postcondition) pairs are respected and which are not. Due to mutual exclusivity of the preconditions, we can deduce which rules *must* be in $I$ from the respected pairs and which rules *must not* be in $I$ from the disrespected pairs. These form the lower bound and upper bound of $I$ and are defined as the insets and outsets, respectively. The function insets_and_outsets therefore enumerates all possible realizations $G'$ (the product over all edge directions for each adjacent neighbor of $V_S$ in $G$), then computes the inset and outset for each realization.

---

**Algorithm 1:** function grammar_induction(dataset)

**Input:** $\mathcal{D} = [(H_i, \lambda_i) \mid i = 1, \ldots, |\mathcal{D}|]$; $\Sigma$; // dataset of DAGs labeled by $\lambda$, vocabulary of labels

1   $N \leftarrow \{\}$; $T \leftarrow \{\text{black}\}$; $P \leftarrow \{\}$
2   $G := (\Sigma, N, T, P, \text{black})$; // initialize grammar
3   $S \leftarrow [[], i \in \{1, \ldots, |\mathcal{D}|\}]$;
4   iter $\leftarrow 0$;
5   **while** len($\mathcal{D}$) $> 0$ **do**
6      $G\_\text{iter} \leftarrow$ learn_grammar($\mathcal{D}$);
7      $G\_\text{iter}, \mathcal{D}, S\_\text{iter} \leftarrow$ disambiguate($G\_\text{iter}, \mathcal{D}$);
8      **for** $i \in S\_\text{iter}$ **do**
9          $S[i] \leftarrow S\_\text{iter}[i]$;
10     **for** $(X, D, I) \in G\_\text{iter}.P$ **do**
11         $G.P \leftarrow G.P \cup \{(X : \text{iter}, D, I)\}$;
12     iter$+= 1$;
13   **while** iter $> 0$ **do**
14     iter$-= 1$;
15     $G.P \leftarrow G.P \cup \{\text{black}, \text{black} : \text{iter}, \{\}\}$; // abbrev: graph with single node labeled black:iter
16   Out: $G, S$

---

## B.3. Find Compatible Isomorphisms

Given a way to compute the insets and outsets for a given subgraph occurrence $S$ and potential edge redirection, we need a way to reconcile different such instances $[S_{i,j} \mid$ subgraph occurrence $i$, redirections $j]$ using their inset and outset. We introduce the notion of a isomorphism compatibility graph, where each node represents a specific occurrence, and edges indicate compatibility, i.e. there exists an instruction set $I$ that is compatible with both. We can define compatibility between $S_{i,j}$ and $S_{i',j'}$ as: "there exists some set $I_{i,j}$ which includes insets of $S_{i,j}$ and $S_{i',j'}$ and excludes outset of $S_{i,j}$ and $S_{i',j'}$", as on line 29 of 3. Given $S$, we also determine whether $S_{i,j}$ should be added to ism_graph for the case $i = j$. Once we have

---

**Algorithm 2:** function learn_grammar(dataset)

---

**Input:** $\mathcal{D} = [(H_i, \lambda_i) \mid i = 1, \ldots, |\mathcal{D}|]; \Sigma;$ // dataset of DAGs labeled by $\lambda$, vocabulary of labels

1   $H =$ disjoint union of $\mathcal{D}$;
2   $N \leftarrow \{\text{gray}, \text{black}\}; T \leftarrow \{\text{black}\}; P \leftarrow \{\}$
3   $G := (\Sigma, N, T, P, \text{black});$ // initialize grammar
4   $M = |H| + 1; t \leftarrow 0;$
5   **while** $|H| < M$ **do**
6      $M \leftarrow |H|;$
7      $m \leftarrow |H| + 1;$
8      **while** $|H| < m$ **do**
9         $m \leftarrow |H|;$
10        best_clique $\leftarrow [];$
11        best $\leftarrow H;$
12        **for** $(X, D, I)$ in $P$ **do**
13           ism_graph $\leftarrow$ find_iso$(H, D, I);$
14           max_clique $\leftarrow$ approx_best_clique(ism_graph);
15           **if** $|\text{max\_clique}| \cdot |D| > |\text{best\_clique}| \cdot |\text{best}|$ **then**
16              best_clique $\leftarrow$ max_clique;
17              best $\leftarrow D;$
18        **for** $d \in$ best_clique **do**
19           rewire$(H, d);$
20      motifs $\leftarrow$ frequent_subgraph_mining$(H);$
21      best_clique $\leftarrow [];$
22      best $\leftarrow H;$
23      **for** $D \in$ motifs **do**
24        ism_graph $\leftarrow$ find_iso$(H, D);$
25        max_clique $\leftarrow$ approx_best_clique(ism_graph);
26        **if** $|\text{max\_clique}| \cdot |D| > |\text{best\_clique}| \cdot |\text{best}|$ **then**
27           best_clique $\leftarrow$ max_clique;
28           best $\leftarrow D;$
29        $I \leftarrow \bigcup_{d \in \text{best\_clique}} d.\text{inset};$
30        $P \leftarrow P \cup \{(\text{gray}, \text{best}, I)\};$
31      **for** $d \in$ best_clique **do**
32        rewire$(H, d);$
33      $t \leftarrow t + 1;$
34   **for** $D \in$ connected_components$(H)$ **do**
35      $P \leftarrow P \cup \{(\text{black}, D, \{\})\};$
36   Out: $G$

---

ism_graph, we extract the maximum clique of this graph, as that maximizes the compression for this given isomorphism equivalence class.

## C. Disambiguation Algorithm and Analysis

### C.1. Pseudocode

In Algo. 4, we give the pseudocode of the disambiguation algorithm.

---

**Algorithm 3:** function find_iso(H,D,I=None)

---

**Input:** $H$; $(D, \lambda_D)$; // `background graph, subgraph`

1   isms $\leftarrow$ fast_subgraph_isomorphism$(H, D)$;
2   term_only $\leftarrow$ all$(\lambda_D(x) \in N, \forall x \in D)$;
3   isms_allowed $\leftarrow []$;
4   **for** ism $\in$ isms **do**
5      $D$_ism, $\lambda_{\text{ism}} \leftarrow$ ism;
6      **if** !term_only **then**
7         isms_allowed += [ism];
8         continue;
9      has_nt $\leftarrow$ any$(\lambda_{\text{ism}}(x) \in N, \forall x \in D$_ism$)$;
10     **if** !has_nt **then**
11        isms_allowed += [ism];
12        continue;
13   $V \leftarrow \{\}; E \leftarrow \{\}$; // `undirected graph`
14   **for** ism $\in$ isms_allowed **do**
15      redirections $\leftarrow$ insets_and_outsets$(H, \text{ism})$;
16      **for** inset, outset, dirs $\in$ redirections **do**
17         **if** $I! = $ None **then**
18            **if** !empty(inset $- I$) **then**
19              continue;
20            **if** !empty(outset $\cap I$) **then**
21              continue;
22         **else**
23            **if** inset $\cap$ outset **then**
24              continue;
25            new_node $\leftarrow \{\text{ins} = \text{inset}, \text{out} = \text{outset}, \text{ism} = \text{ism}, \text{dirs} = \text{dirs}\}$;
26            $V \leftarrow V \cup \{\text{new\_node}\}$;
27   **for** $i \in V$ **do**
28      **for** $j \in V$ **do**
29         overlap $\leftarrow (i.\text{inset} \cup j.\text{inset}) \cap (i.\text{outset} \cup j.\text{outset})$;
30         **if** !overlap **then**
31            $E \leftarrow E \cup \{(i, j)\}$;
32      Out: $V, E$

---

### C.2. Proof of Correctness

**Lemma.** The output $G$ of Algorithm 4 is unambiguous w.r.t. $\mathcal{D} \cap L(G)$.

**Proof.** To see this, we work backwards from the definition of minimal_rule_set_selection, which is assumed to solve the problem in Theorem C.3. Therefore, elim_rules will be a superset of at least one element in elim_rule_sets for each $i$. Each element of elim_rule_sets is a set consisting of all rules which should be eliminated to ensure $H_i$ becomes unambiguous. This is ensured by construction because for each derivation whose set of rules is unique, we try excluding all other derivations. Consider two derivations A and B with rule sets set(A) and set(B), where we want to keep A valid but invalidate B. Then, we can eliminate rules set$(B) \setminus$ set$(A)$. This is possible because there does not exist two derivations where one's rule set is a subset of the other's rule set. This is because each rule application adds a positive number of nodes, since the RHS of any rule contains at least two nodes. Therefore, we construct elim_sets, a set of rule set differences for keeping keep_deriv. We then find a hitting set of elim_sets (rules which invalidates other derivations but keeps the current derivation valid). Therefore, the solution from minimal_rule_set_selection will be the minimal set of rules which disambiguates all $H_i$ which

can be made unambiguous.

**Theorem.** The output $G$ of Algorithm 1 is unambiguous w.r.t. $\mathcal{D}$.

**Proof.** Algorithm 1 constructs a compound grammar which make consist of multiple sub-grammars that each guarantees unambiguity for a partition of $\mathcal{D}$. Each sub-grammar's non-terminals are identified by iter so any derivation over $G$ stays strictly within one sub-grammar. Thus, showing $\not\exists H_i \in \mathcal{D}$ s.t. $H_i$ is ambiguous w.r.t. $G$, reduces to showing $\not\exists H_i \in L(G\_\text{iter}) \cap \mathcal{D}$ which is ambiguous w.r.t. $G\_\text{iter}$ for a given iteration.

### C.3. Undecidability of Detecting Ambiguity

**Theorem.** Given edNCE grammar $G = (\Sigma, N, T, P, S)$, testing if it is ambiguous is undecidable.

**Proof.** Suppose determining whether $G$ is ambiguous is decidable. Then we can reduce determining whether a string grammar is ambiguous is decidable by reducing it to an equivalent edNCE grammar (Engelfriet & Rozenberg, 1997). However, determining whether a string grammar is undecidable (Brabrand et al., 2010), which is a contradiction.

### C.4. Formulation of Disambiguation

**Theorem.** Given a universe $U$ and a collection of *sets of subsets*, $\mathcal{S} = \{S_1, S_2, \ldots, S_M\}$, $S_i \in 2^{2^U}$. Let $k$ be an integer. Determining whether $\exists H \subseteq U$ such that $|H| \leq k$ and

$$\forall i \in \{1, 2, \ldots, M\}, \exists T \in S_i \text{ s.t. } T \subseteq H \tag{1}$$

is NP-complete.

**Proof.** Let $HSS := \{(U, \mathcal{S}, k)\}$ s.t. $|H| \leq k$ and 1 is satisfied.

**$HSS$ is in NP**: For each $S_i \in \mathcal{S}$, non-deterministically guess a $T_i \in S_i$. Let $H := \bigcup_i T_i$. If $|H| \leq k$, accept, else reject.

**$HSS$ is NP-hard**: Let $HS := \{(U, \mathcal{S}, k)\}$ s.t. $\exists H \subseteq U$ s.t. $|H| \leq k$ and $H \cap S_i \neq \emptyset$ for every $S_i \in \mathcal{S}$ be the Hitting Set problem, which is known to be NP-complete. We will show $HS \leq_m HSS$. Given an instance of the HS problem, let $f$ be the computable mapping $f((U, \mathcal{S}, k)) = (U, \{\{\{s\}, \forall s \in S_i\}, \forall S_i \in \mathcal{S}\}, k)$. If $(U, \mathcal{S}, k) \in HS$, then we can choose $s_i \in S_i \in H \cap S_i$ to be $T_i$ for each $i$. Then we can choose $H' = \{s_i\}$ so 1 is satisfied by construction, and since $s_i \in H$ for each $i$, $H' \subseteq H$ so $|H'| \leq k$. If $f((U, \mathcal{S}, k)) \in HSS$, then let $H'$ satisfy $|H'| \leq k$ and 1. Then $T_i \subseteq H'$ where $|T_i| = 1$ is equivalent to $\exists s_i \in H'$ for each $i$, or $H' \cap S_i \neq \emptyset$, so since $|H'| \leq k$, $(U, \mathcal{S}, k) \in HS$.

**Corollary.** The problem minimal\_rule\_set\_selection is solving is NP-Complete.

## D. Grammar Enumeration Algorithm

It is well-known node-labeled DAGs can be hashed by recursively aggregating hashes of children. We use a simple approach in our implementation (Algo. 5). Note that edge-labeled DAGs can be polynomial-time reduced to node-labeled DAGs, so our approach works in the general edNCE case. For a recent discussion of hashing directed graphs, refer to Helbling (2020).

### D.1. Dynamic Programming with Memoization.

We use memoization to make the brute force enumeration tractable, along with efficient pruning. In our implementation (Algo. 7), intermediate derivations are pruned if a) they are not DAGs, or b) are not node-induced subgraphs of the desired DAG. mem stores all derivations "to-go" for a given intermediate, so the given intermediate is memoized.

### D.2. Computational Efficiency.

The worse-case complexity is, in the general case, NP-hard, because parsing edNCE grammars are NP-hard (Engelfriet & Rozenberg, 1997). Intuitively, there can be an *exponential* number of connected subgraphs for a given DAG (tight for the case of star graphs), though isomorphisms and sparsity means the actual number is lower. In our practical experiments of path-like structures, the algorithm is very efficient (the multi-process version of Algo. 7 takes a few minutes per DAG for BN and CKT). For ENAS, the much larger number of rules creates a large branching factor, but the sparser, path-like structures enables more pruning, still making the algorithm tractable. We also run Algo. 7 in order from the smallest to largest DAGs, as smaller DAGs likely have shorter derivations that enable more rules to be pruned before enumerating

derivations for the larger DAGs.

## D.3. Remarks on Scalability

Our datasets BN, CKT, ENAS are all upper-bounded in the number of nodes, which makes the brute force approach tractable. Further optimizations are required for variable-size DAGs, where domain knowledge can further prune intermediates.

In cases where brute-force approaches are not feasible, we have the following suggestions:

1. If the issue lies in the large $|\mathcal{D}|$, we suggest partitioning $\mathcal{D}$ based on some semantic criterion, then running Algo. 1 on those individual partitions, then aggregating the individual grammars into a compound grammar much like how we did for Algo. 1. The drawback is this pre-partitioning scheme loses the injectivity property when viewing $\mathcal{D}$ as a whole, but retains injectivity for the individual "sub-datasets".

2. In cases where individual graphs in $\mathcal{D}$ are too large, we suggest increasing the "motif size" for Subdue, as larger candidate motifs produce shorter derivations. The ideal derivation length is somewhere between 2-8, in our empirical experience. The drawback is this may result in lower compression ratios, depending on the characteristics of the data. We encourage future work to explore this further.

## E. Case Study: Analog Circuits

**Background.** Operational amplifiers (op-amps) are a DAG generation and optimization problem because their circuit topologies inherently form DAG structures. The design of op-amps involves both topology selection and device parameter optimization, making it a highly complex, combinatorial problem. Traditionally, op-amp optimization has focused primarily on device sizing (component-wise parameters) given a fixed topology, but recently graph generative models have shown promise in optimizing the DAG topology. (Dong et al., 2023) However, general methods that navigate the combinatorial search space without domain-specifc knowledge is challenging, whereas specialized methods will not generalize to other problem domains. For example, Dong et al. (2023) used a two-level GNN on top of a predefined basis of circuit subgraphs, facilitating domain-specific representation learning. DIGGED, by contrast, combines the flexbility of a general method with data-driven grammar induction. Essentially, DIGGED infers the expert knowledge indirectly, through its unsupervised MDL objective.

### E.1. Case Study Example

We visualize the novel design with highest simulated FoM generated by DIGGED during BO in Fig. 6. Shown in 6a, DIGGED derives this design by decoding three tokens. In the first step, it decodes one of the common initial rules to initialize the input and output, leaving the middle as a placeholder. In the second step, it decodes a rule which adds a resistor, a stabilizing mechanism for the yet-to-be decoded structure. In the final step, it decodes the rule which contains two -gm+ op-amp stages. This is interesting, because the final token decoded is inspired by its previous token. Using a parallel resistor configuration is one of the common ways to provide stability to a two-stage op-amp. In Fig. 6c, we visualize the instruction set $I$ for Rule 56, which controls what neighborhood topology should surround the two -gm+ cells. This instruction set is the solution to the compatibility maximization (Section B), so it contains some redundancy in the context of this specific example. For example, both -gm+ cells in Fig. 6c have preconditions for connecting to input and output nodes, but in the context of any specific derivation, at most one will be active. However, *only* the upper -gm+ cell has preconditions for resistor, capacitor and other gm nodes. This captures important constraints, notably: we don't want other gm cells to connect to both -gm+ stages, because we want cascaded gain blocks and sequential separation of the units. This is significant from a methodology perspective because DIGGED is inducing symbolic rules such as these directly from examples, so it won't construe unintended topology, whereas other autoregressive graph decoders might. Furthermore, these step-by-step derivations provide **explainability** into the designs, whereas decoding a DAG in an arbitrary order might miss this information.

### E.2. Expert Feedback

We visualize the four novel designs with highest FoMs in Figs. 7a-7c. We consulted an expert with decades of experience in circuit design, and include the feedback in the captions.

### E.3. More Details on Baselines

Similar to the setup for ENAS and BN Zhang et al. (2019); Thost & Chen (2021), we retrain the SGP model each round and acquire latent points using the Greedy Expected Improvement heuristic. For each latent point, we decode a DAG using the decoder and convert it to a circuit. We refer to this as *unconstrained* BO, and adapt the existing implementations by Zhang et al. (2019); Thost & Chen (2021). We also include their reported BO results in Table 3. However, we could not find support for this in the codebase of Dong et al. (2023). Instead, they provide instructions to run BO with *pivots*, where they first generate latent encodings of all circuits in their benchmark dataset, CktBench301, then snap each acquisition point to the closest circuit in the dataset. Thus, it's unclear how they obtained the numbers in their Table 1. For completeness, we ran their code and include the results as CktGNN [CktBench301] in Table 3. Despite using a large enumerated dataset as pivots, they were unable to produce designs close to the max FoM in CktBench301.

### E.4. Summary

DIGGED demonstrates, through a domain-agnostic and unsupervised paradigm, it is capable of achieving greater performances than domain-specific methods. It does so by autonomously discovering domain-specific patterns, automatically inducing principled and compact sequential descriptions over those patterns, and harnessing general-purpose sequence learning.

## F. Case Study: Bayesian Networks

In Section 5.2, we observed an interesting finding, where DIGGED achieves extraordinary Pearson $r$ (nearly 1.6x that of the next best method and 2.9x that of other VAE encoders) despite a modest RMSE. To understand this phenomenon, we visualize the trained SGP's test set predictions (one of the ten seeds) in Fig 8. We then stratify the test set across the number of rules in the sequence representation, and plot the mean absolute error per strata.

**Findings.** We indeed see a tight, linear trend of the predictions across the entire ground-truth value range in Fig. 8, despite the error residues being high. We also see this test error is highest for examples with parse length 1. Fortunately, there are very few of those. We then observe a generally decreasing trend in the test error as parse length increases. Notably, 2 and 3 are the most common parse lengths, but exhibits relatively low test errors. We attribute the linear trend to the unique properties of our representation. In contrast with node-by-node or edge-by-edge sequential decoding schemes, DIGGED uses compatible and consistent rules to linearize a DAG. This reparameterizes the DAG representation space into a sequence representation space, where Transformers have shown strong generalization abilties (Vaswani, 2017). Thus, although individual residue errors are still large, a global linear trend emerges from this representation space. This shows how theoretical properties of our grammar translate into more congruous representations that are amenable for downstream tasks.

## G. Further Discussion of Results

**Relation to Compositional Generalization.** In addition to recognizing our method as "converting a graph into a sequence", there is a deep motivation from the objectives of compositional generalization. Contrary to what a "sequential" representation may imply, DIGGED is intrinsically compositional (see 2). In fact, DIGGED is trained to embed DAGs with similar hierarchical compositions to similar points in latent space. For example, consider DAG 1 represented (uniquely) as $W \rightarrow X \rightarrow Y$ and DAG 2 as $W \rightarrow X \rightarrow Z$. DIGGED's decoder must predict shared initial tokens for both graphs, naturally clustering these related graphs in latent space. Combined with the relational inductive bias of our DAGNN encoder, the autoencoder objective can be viewed as combining both the relational and hierarchical inductive bias to learn expressive and generalizable representations.

**How Choice of Datasets Affect Interpretation of Results**. ENAS and BN both impose special constraints: all DAGs have the same number of nodes; ENAS DAGs must follow consecutively numbered nodes, and BN DAGs must contain exactly one node of each type (8 types). Such simplifying conditions allow naïve positional encodings to overcome the shortcomings we discussed earlier, making predictive tasks relatively easier. We initially chose these datasets due to the limited availability of standardized benchmarks for DAGs. By contrast, the CKT dataset involves significant diversity in both graph topology and node types, making it a better testbed for evaluating the true strengths of DIGGED's compositional, position-free encoding approach. At the same time, we note predictive accuracy (RMSE, Pearson's r) does not reflect decoder effectiveness. For example, BN-Random, CKT-BFS, and CKT-Random achieve reasonable scores on predictive

metrics (RMSE and Pearson's r), yet fail fundamental decoder sanity checks, rendering them ineffective for subsequent optimization tasks. DIGGED prioritizes end-to-end optimization results, which requires the ability to navigate and decode from the latent space.

**Explanation for Better Optimization, Worse Predictive Accuracy.** The opportunities and challenges of hierarchical, compositional generalization also explains the behavior we observe in Ablation 6.1. DIGGED is intentionally designed for compositionality of its outputs. Unlike naive sequential encodings, DIGGED places DAGs with shared hierarchical structures (intermediate derivations) close together in the latent space. Learning both the token vocabulary embeddings and latent space compositional structure jointly indeed poses a more challenging training task – reflected partly in predictive metrics – but strongly supports compositional generalization and decoder reliability. This trade-off underscores DIGGED's core strength: effectively navigating a compositional design space to reliably generate diverse and valid DAG structures optimized for practical performance. DIGGED is also a design language, combining hierarchical inductive biases and can uncover domain-specific insights (case studies in App. E).

## H. Choice of Encoder

In Fig. 9, we plot the frequency of rule tokens, sorted by rank (from most to least common) against Zipf's Law (Zipf, 2013), a cornerstone of modern linguistics. Zipf's Law states that the frequency of the $k$'th most common word is inversely proportional to its rank, and this arises in many natural settings (Powers, 1998). It's encouraging to see that our unsupervised MDL-based compression scheme also gives rise to such an underlying relationship. Similar to natural language, we believe the formal language behind DIGGED also shares similar governing laws, which would be fascinating to study in its own right.

Despite the theory and intuitions, transferring modern practices in NLP directly onto our framework did not strike gold on the first try. We showed in our ablations in Section 5 that using a full transformer encoder (TOKEN) did worse than using a GNN tailored to the inductive biases of DAGs. We postulate two reasons for this:

1. **Transformer encoders require more investment in training.** This is supported by our hyperparameter experiments in I, where we noticed the encoder required twice as many layers as our decoder. Because the focus of our work is not pretraining, we did not invest the time to pretrain the encoder separately.

2. **Jointly training an encoder, decoder, and dictionary is data-intensive**. For this reason, pretrained word embeddings (Mikolov et al., 2013; Pennington et al., 2014) are used out-of-the-box for joint encoder-decoder training (Vaswani, 2017; Raffel et al., 2020). However, distributed embeddings for our rule-based tokens do not exist. However, since we are, to our knowledge, among the first to train generative models by representing graphs as sequences of tokens, such solutions do not currently exist.

We believe standard encoder pretraining practices like masked language modeling (Devlin, 2018) will be effective. We encourage future works to explore this direction further with larger datasets, and we believe there will be scaling laws akin to those we have seen in modern language models (Kaplan et al., 2020). We also encourage future works to explore distributed embeddings by viewing graphs as documents and neighborhood topologies as context windows. Another promising way to bootstrap token embeddings is to leverage the inductive bias of its definition (daughter graph $D$ and instruction set $I$). We hope our work opens the Pandora's Box of graph language modeling using lossless, sequential descriptions!

## I. Hyperparameter Scan

The optimal parameters for our model were determined using a hyperparameter scan sweeping over various properties of the VAE, using validation loss as the guide. During the scan, we explore varying architecture properties such as: number of encoder layers, number of decoder layers, latent dimension, embedding dimension, batch size, and KL divergence loss coefficient. We employed the validation loss of the VAE to guide parameter selection, updating one hyperparamter at a time while keeping all others fixed at baseline values. After each scan, we locked in the best-performing setting before moving on to the next parameter type. The ordering and description of each hyperparameter that was optimized is as follows. We also include the default setting of each parameter in parenthesis:

1. *Number of Decoder Layers:* Depth of the Transformer decoder. **(4)**

2. *Number of Encoder Layers:* Depth of the Transformer encoder. **(4)**

3. *KL Divergence Loss Coefficient:* Scalar coefficient of the KL divergence term in the typical VAE loss function (Evidence Lower Bound, ELBO). Controls how closely the encoder's latent distribution matches the prior. **(0.5)**

4. *Batch Size:* Number of training examples processed simultaneously for a gradient update. **(256)**

5. *Latent Dimension:* Size of the representation of input sequences in the latent space of the variational autoencoder. **(256)**

6. *Embedding Dimension:* Size of the embeddings that the encoder and decoder use to represent tokens. **(256)**

The chosen parameters values from each experiment are highlighted in green in Table 8 and Table 9.

Interestingly, for the "Sequence Rule" encoder on the CKT dataset, we achieve the lowest validation loss with just 4 Transformer decoder layers, whereas the Transformer encoder requires 8. This shows that DIGGED works well with a lightweight decoder. We attribute this to the compactness of the grammar. Given a good sequential description, decoding can be streamlined significantly.

It is worth noting that due to time and resource constraints, we were only able to fully scan the hyperparameters for a subset of the possible encoder-type—dataset combinations.

---

**Algorithm 4:** function disambiguate(G,S)

**Input:** $G; \mathcal{D}$ // `learned grammar, dataset`

1   all_elim_rule_sets $\leftarrow \{\}$;
2   all_derivs $\leftarrow []$;
3   **for** $(H_i, \lambda_i) \in \mathcal{D}$ **do**
4      derivs $\leftarrow$ enumerate_derivations($H_i$);
5      deriv_rule_set_lookup $= \{\}$;
6      **for** deriv $\in$ derivs **do**
7          key $\leftarrow$ sorted(list(set(deriv)));
8          deriv_rule_set_lookup[key]$+ =$ [deriv];
9      umabig_poss $\leftarrow$ False;
10      **for** key $\in$ deriv_rule_set_lookup **do**
11          **if** deriv_rule_set_lookup[key] $== 1$ **then**
12              umabig_poss $\leftarrow$ True;
13              break;
14      **if** !umabig_poss **then**
         // `impossible to make unambiguous, later will be lost`
15          all_derivs $\leftarrow$ umabig_poss $+ [[]]$;
16          continue;
17      all_derivs $\leftarrow$ all_derivs $+$ [derivs];
18      elim_rule_sets $\leftarrow \{\}$;
19      **for** key $\in$ deriv_rule_set_lookup **do**
20          **if** len(deriv_rule_set_lookup[key]) $> 1$ **then**
21              continue;
22          keep_deriv $\leftarrow$ deriv_rule_set_lookup[key][0];
23          elim_sets $\leftarrow \{\}$;
24          **for** deriv $\in$ derivs **do**
25              **if** deriv $==$ keep_deriv **then**
26                  continue;
27              elim_sets $\leftarrow$ elim_sets $\cup$ {deriv) \ set(keep_deriv)};
28          elim_rule_set $\leftarrow$ quick_hitting_set(elim_sets);// `inner hitting set problem`
         // `we use a linear greedy implementation`
29          elim_rule_sets $\leftarrow$ elim_rule_sets $\cup$ {elim_rule_set};
30      all_elim_rule_sets $\leftarrow$ all_elim_rule_sets $\cup$ {elim_rule_sets};
31 elim_rules $\leftarrow$ minimal_rule_set_selection(all_elim_rule_sets);// `given a set of set of subsets, find`
     `minimal rules to eliminate so each set of subsets has at least one subset included`
32 $G.P \leftarrow G.P \setminus$ elim_rules;
33 dataset $\leftarrow []$;
34 unique_derivs $\leftarrow \{\}$;
35 **for** $(H_i, \lambda_i) \in \mathcal{D}$ **do**
36      lost $\leftarrow$ True;
37      **for** deriv $\in$ all_derivs[$i$] **do**
38          **if** empty(set(deriv) $\cap$ elim_rules) **then**
39              lost $\leftarrow$ False;
40              unique_derivs[$i$] $\leftarrow$ deriv;
41              break;
42      **if** lost **then**
43          dataset $\leftarrow$ dataset $+ [(H_i, \lambda_i)]$;
44 Out: $G$, dataset, unique_derivs

---

---

**Algorithm 5:** function wl_hash(H)

---

**Input:** $H$; // `DAG`

1  $G \leftarrow \text{deepcopy}(H)$;
2  $G \leftarrow \text{relabel\_nodes}(G, \text{dict}(\text{zip}(\text{sorted}(G.\text{nodes}()), \text{range}(|G|))))$;
3  $m \leftarrow |G.\text{edges}|$;
4  $\text{edge\_index} \leftarrow \emptyset^{2 \times m}$;
5  $\text{edge\_index}[:, : m] \leftarrow \text{array}(G.\text{edges}).T$;
6  $\text{roots} \leftarrow \text{setdiff}(\{0, \ldots, |G| - 1\}, \text{edge\_index}[1])$; // `Nodes with no predecessors`
7  $\text{colors} \leftarrow \{\}$;
8  **for** $r \in \text{roots}$ **do**
9  $\quad$ wl_hash_node$(G, r, \text{colors})$;
10 $\text{ans} \leftarrow \text{`—'}.\text{join}(\text{sorted}([\text{colors}[r] \,|\, r \in \text{roots}]))$;
11 $\text{hash\_value} \leftarrow \text{sha256}(\text{ans.encode}()).\text{hexdigest}()$;
12 Out: hash_value

---

**Algorithm 6:** function wl_hash_node(G, n, colors)

---

**Input:** $G$; $n$; colors

1  **if** $n \in \text{colors}$ **then**
2  $\quad$ Out: colors$[n]$;
3  **if** $G[n] \neq \emptyset$ **then**
4  $\quad$ $\text{cs} \leftarrow \text{sorted}([\text{wl\_hash\_node}(G, c, \text{colors}) \,|\, c \in G[n]])$;
5  $\quad$ $\text{val} \leftarrow G.\text{nodes}[n][\text{`label'}])$; // `symbol in` $N \cup T$
6  $\quad$ $\text{val} \leftarrow \text{val} + \text{`,'} + \text{` '}.\text{join}(\text{cs})$;
7  **else**
8  $\quad$ $\text{val} \leftarrow G.\text{nodes}[n][\text{`label'}])$;
9  $\text{colors}[n] \leftarrow \text{val}$;
10 Out: colors$[n]$

---

**Algorithm 7:** function enumerate_derivations(index, all_derivs, grammar, graph)

---

**Input:** index; all_derivs; grammar; graph

1  **if** index $\in$ all_derivs **then**
2  $\quad$ log(f"index enumerated");
3  $\quad$ Out: all_derivs[index];
4  $G \leftarrow \text{DiGraph}()$;
5  $G.\text{add\_node}(\text{`0'}, \text{label} = \text{`black'})$;
6  $\text{init\_hash} \leftarrow \text{wl\_hash}(G)$;
7  $\text{stack} \leftarrow [(\text{deepcopy}(G), \text{init\_hash})]$;
8  $\text{mem} \leftarrow \{\}$;
9  **while** stack $\neq \emptyset$ **do**
10 $\quad$ worker_single(stack, grammar, graph, init_hash, mem); // `Here we use multi-processing, omitted for`
$\quad\quad$ `simplicity`
11 $\text{derivs} \leftarrow \text{mem}[\text{init\_hash}]$;
12 $\text{all\_derivs}[\text{index}] \leftarrow \text{derivs}$;
13 Out: derivs

---

---

**Algorithm 8:** function worker_single(stack, grammar, graph, init_hash, mem)

---

**Input:** stack; grammar; graph; init_hash; mem

1 **while** True **do**
2    **if** stack $= \emptyset$ **then**
3       **if** init_hash $\in$ mem $\wedge$ mem[init_hash] $\neq 0$ **then**
4          break;
5    $(H, \mathrm{val}) \leftarrow$ stack.pop();
6    **if** val $\in$ mem **then**
7       **if** mem[val] $\neq 0$ **then**
8          continue;
9    **else**
10       mem[val] $\leftarrow 0$;
11    nts $\leftarrow \{v \in H \mid v \in N\}$;
12    **if** nts $= \emptyset$ **then**
13       **if** is_isomorphic$(H, \mathrm{graph}, \mathrm{node\_match})$ **then**
14          mem[val] $\leftarrow [[]]$;
15       **else**
16          mem[val] $\leftarrow []$;
17       continue;
18    done $\leftarrow$ True; res $\leftarrow []$;
19    **for** nt $\in$ nts **do**
20       **for** rule $\in$ grammar.rules **do**
21          **if** rule $=$ None **then**
22             continue;
23          nt_label $\leftarrow H_V[\mathrm{nt}]['\mathrm{label}']$;
24          **if** rule.nt $=$ nt_label **then**
25             c $\leftarrow$ rule$(H, \mathrm{nt})$;
26             **if** $\neg$is_connected(Graph(c)) **then** continue; ;
27             **if** $\neg$is_directed_acyclic_graph(c) **then** continue; ;
28             **if** $\neg$find_partial([graph], c) **then** continue; ;
29             hash_val $\leftarrow$ wl_hash(c);
30             **if** hash_val $\notin$ mem **then**
31                **if** done **then**
32                   stack.append$((H, \mathrm{val}))$;
33                   done $\leftarrow$ False;
34                stack.append$((c, \mathrm{hash\_val}))$;
35             **else**
36                **if** mem[hash_val] $= 0$ **then**
37                   **if** done **then**
38                      stack.append$((H, \mathrm{val}))$;
39                      done $\leftarrow$ False;
40                **else**
41                   **for** seq $\in$ mem[hash_val] **do**
42                      res.append([i] + deepcopy(seq));
43    **if** done **then**
44       mem[val] $\leftarrow$ res;

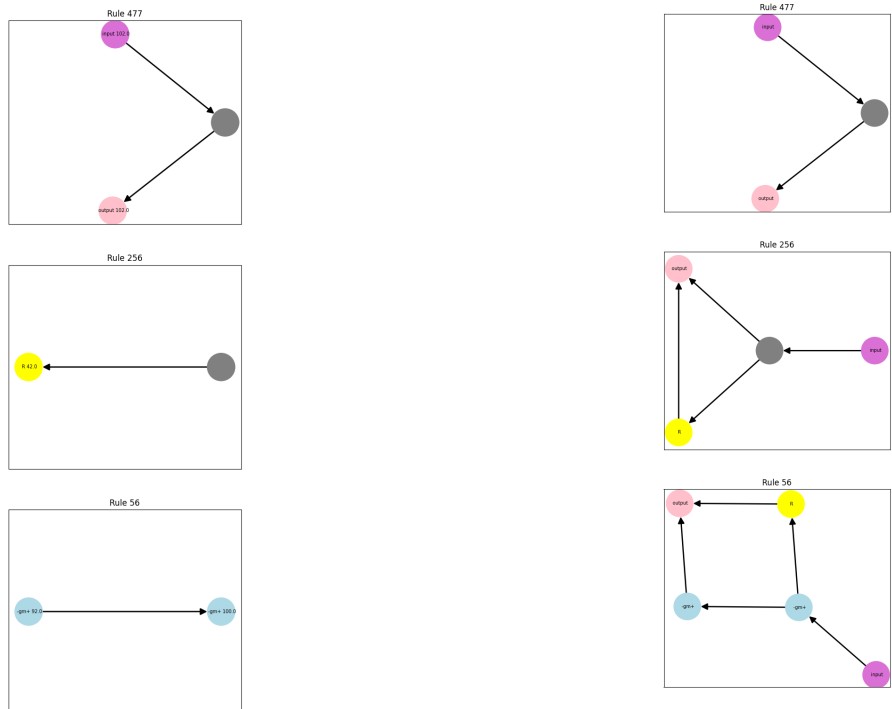

(a) Parse (top-to-bottom) representation.

(b) Step-by-step derivation for the design (not shown: instruction set per rule).

## RHS

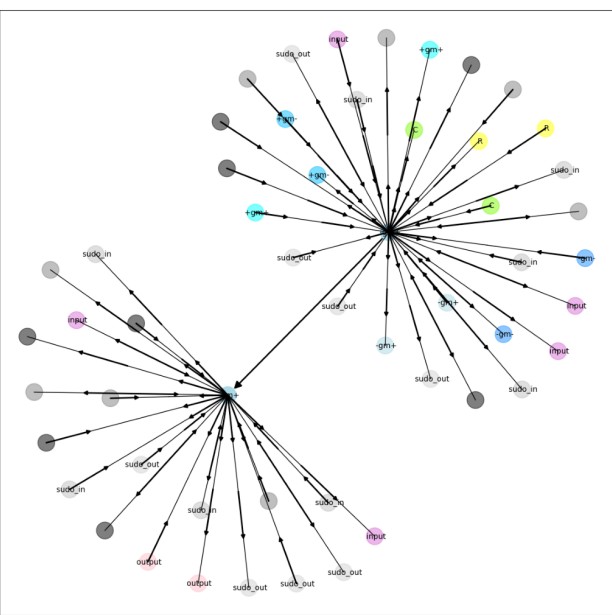

(c) Visualization of instructions for rule 56. The daughter graph $D$ consists of two -gm+ cells units (with different device placement parameters). We fan out individual instructions, where we use custom half-arrows to visualize redirections $(d/d')$. For what each element in the tuple means, see Section 3.1.

*Figure 6.* Visualization of case study for the best novel design in Fig. 3.

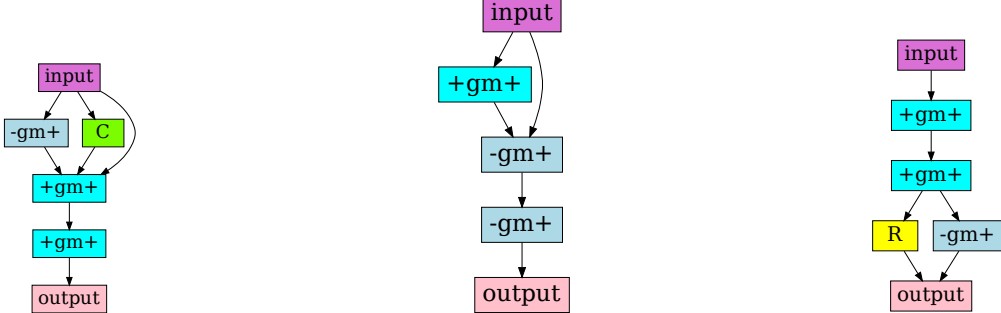

(a) FoM= 265.53, "is a possible circuit topology" - Expert

(b) FoM= 296.82, "a bit fishy, because +gm+ is in parallel with an edge" - Expert

(c) FoM= 243.72, "could be a good design for certain applications" - Expert

*Figure 7.* We include three additional novel designs found during BO. For each, we include a comment by a circuit design expert.

*Table 8.* Hyperparameter scan with "Sequence Rule" encoder type on the CKT dataset. Note that the order of parameter optimization follows the ordering detailed in the text above (left → right and top → bottom).

| Run # | # dec. layers | Validation loss |
|---|---|---|
| 1 | 1 | 4.000 |
| 2 | 2 | 3.919 |
| 3 | 3 | 3.942 |
| 4 | 4 | 3.882 |
| 5 | 5 | 3.909 |
| 6 | 6 | 3.942 |
| 7 | 7 | 3.897 |
| 8 | 8 | 3.917 |

| Run # | # enc. layers | Validation loss |
|---|---|---|
| 1 | 1 | 4.017 |
| 2 | 2 | 3.935 |
| 3 | 3 | 3.882 |
| 4 | 4 | 3.885 |
| 5 | 5 | 3.888 |
| 6 | 6 | 3.908 |
| 7 | 7 | 3.889 |
| 8 | 8 | 3.874 |
| 9 | 9 | 3.882 |
| 10 | 10 | 3.888 |
| 11 | 11 | 3.883 |
| 12 | 12 | 3.874 |
| 13 | 13 | 3.884 |
| 14 | 14 | 3.879 |
| 15 | 15 | 3.894 |
| 16 | 16 | 3.876 |

| Run # | KL Div. coefficient | Validation loss |
|---|---|---|
| 1 | 0.1 | 3.911 |
| 2 | 0.2 | 3.875 |
| 3 | 0.3 | 3.865 |
| 4 | 0.4 | 3.881 |
| 5 | 0.5 | 3.894 |
| 6 | 0.6 | 3.871 |
| 7 | 0.7 | 3.891 |
| 8 | 0.8 | 3.893 |
| 9 | 0.9 | 3.899 |
| 10 | 1.0 | 3.893 |
| 11 | 1.1 | 3.884 |
| 12 | 1.2 | 3.885 |
| 13 | 1.3 | 3.909 |
| 14 | 1.4 | 3.908 |
| 15 | 1.5 | 3.891 |

| Run # | Batch size | Validation loss |
|---|---|---|
| 1 | 16 | 3.959 |
| 2 | 32 | 3.940 |
| 3 | 64 | 3.896 |
| 4 | 128 | 3.919 |
| 5 | 256 | 3.863 |
| 6 | 512 | 3.882 |
| 7 | 1024 | 3.844 |
| 8 | 2048 | 3.851 |

| Run # | Latent dim. | Validation loss |
|---|---|---|
| 1 | 32 | 3.862 |
| 2 | 64 | 3.858 |
| 3 | 128 | 3.862 |
| 4 | 256 | 3.844 |
| 5 | 512 | 3.873 |
| 6 | 1024 | 3.914 |

| Run # | Embedding dim. | Validation loss |
|---|---|---|
| 1 | 32 | 3.957 |
| 2 | 64 | 3.910 |
| 3 | 128 | 3.862 |
| 4 | 256 | 3.844 |
| 5 | 512 | 3.851 |
| 6 | 1024 | 3.872 |

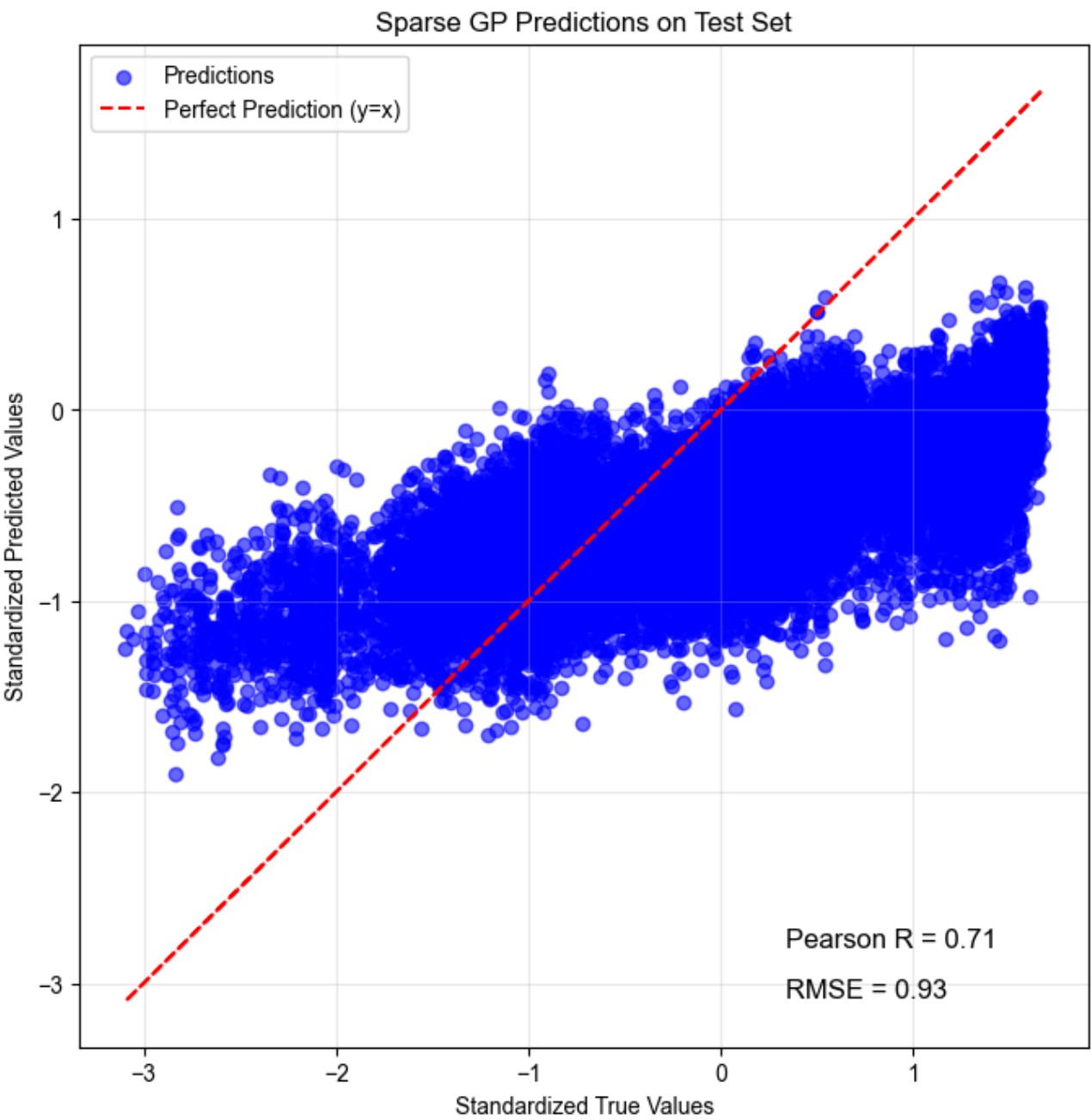

*Figure 8.* We visualize test set predictions of a trained SGP model against the ground-truth.

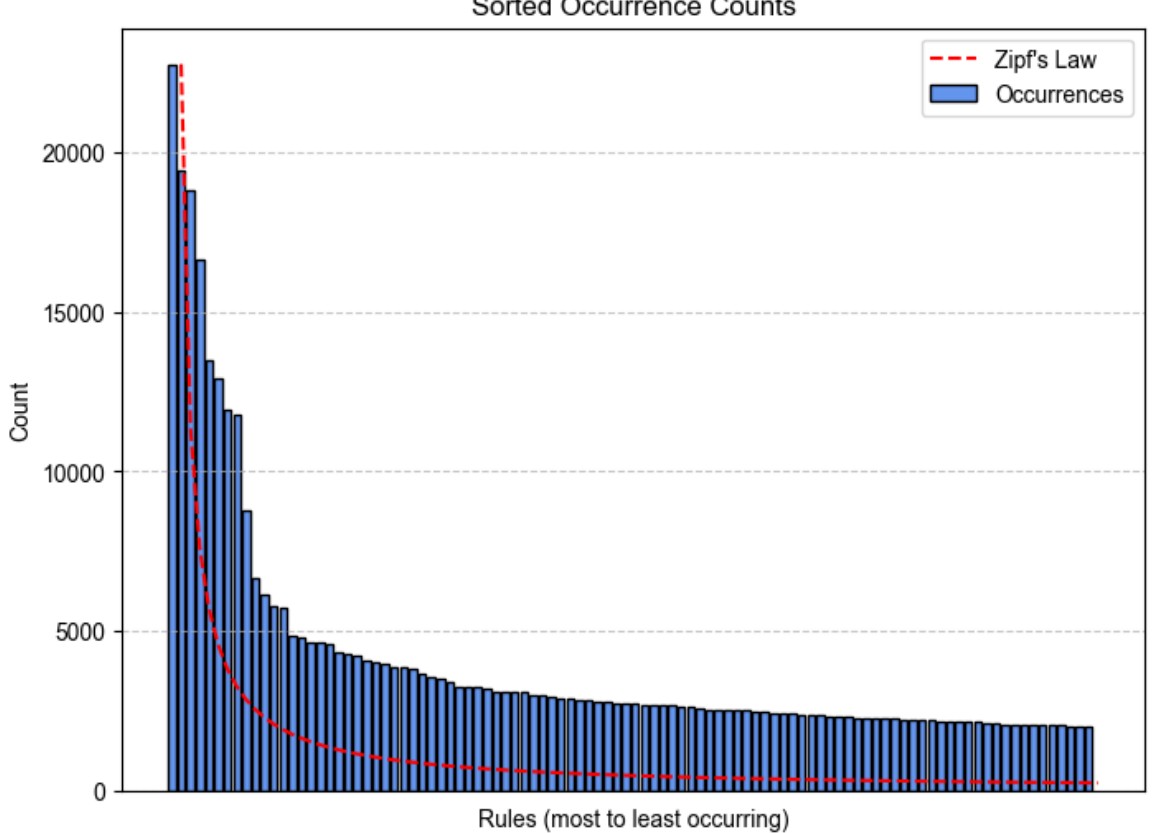

*Figure 9.* We sort all rule tokens by the frequency of occurrence across all sequential descriptions in the BN dataset, benchmarked by Zipf's Law.

*Table 9.* Hyperparameter scan with "Graph" encoder type on the CKT dataset. Note that the order of parameter optimization follows the ordering detailed in the text above (left → right and top → bottom).

| Run # | # dec. layers | Validation loss |
|---|---|---|
| 1 | 1 | 4.014 |
| 2 | 2 | 3.932 |
| 3 | 3 | 3.936 |
| 4 | 4 | 3.937 |
| 5 | 5 | 3.930 |
| 6 | 6 | 3.894 |
| 7 | 7 | 3.915 |
| 8 | 8 | 3.965 |

| Run # | # enc. layers | Validation loss |
|---|---|---|
| 1 | 1 | 3.919 |
| 2 | 2 | 3.925 |
| 3 | 3 | 3.937 |
| 4 | 4 | 3.894 |
| 5 | 5 | 3.924 |
| 6 | 6 | 3.918 |
| 7 | 7 | 3.936 |
| 8 | 8 | 3.939 |

| Run # | KL Div. coefficient | Validation loss |
|---|---|---|
| 1 | 0.1 | 3.919 |
| 2 | 0.2 | 3.900 |
| 3 | 0.3 | 3.901 |
| 4 | 0.4 | 3.965 |
| 5 | 0.5 | 3.919 |
| 6 | 0.6 | 3.951 |
| 7 | 0.7 | 3.959 |
| 8 | 0.8 | 3.949 |
| 9 | 0.9 | 3.979 |
| 10 | 1.0 | 3.988 |

| Run # | Batch size | Validation loss |
|---|---|---|
| 1 | 16 | 3.981 |
| 2 | 32 | 3.912 |
| 3 | 64 | 3.926 |
| 4 | 128 | 3.900 |
| 5 | 256 | 3.900 |
| 6 | 512 | 3.882 |
| 7 | 1024 | 3.883 |

| Run # | Latent dim. | Validation loss |
|---|---|---|
| 1 | 32 | 3.962 |
| 2 | 64 | 3.915 |
| 3 | 128 | 3.898 |
| 4 | 256 | 3.882 |
| 5 | 512 | 3.900 |
| 6 | 1024 | 4.405 |

| Run # | Embedding dim. | Validation loss |
|---|---|---|
| 1 | 32 | 3.957 |
| 2 | 64 | 3.918 |
| 3 | 128 | 3.894 |
| 4 | 256 | 3.882 |
| 5 | 512 | 3.913 |
| 6 | 1024 | 3.908 |

