# OpenReview forum: "Directed Graph Grammars for Sequence-based Learning"
_ICML.cc/2025/Conference — ICML 2025 poster_

### Official Review · Reviewer_7hpe · 2025-03-11

**Overall Recommendation:** 3

**Summary:**

In this paper, the authors propose a novel framework for mapping Directed Acyclic Graphs (DAGs) to sequences via directed graph context-free grammars. Importantly, the authors formulate graph grammar induction as a Minimum Description Length (MDL)-based compression, achieved through an intricate sequence of graph optimization problems. The obtained grammar enables subsequent sequence-based learning tasks using transformers (possibly opening a new bridge between graph-based problems and the emerging LLM technologies) and variational autoencoders (VAEs). Empirical results are demonstrated on small- to medium-scale datasets involving neural architectures, Bayesian networks, and circuit designs.

## Update after rebuttal
I believe the clarifications and additional experiments promised by the authors constitute a good improvement of the present manuscript. While I still find the presentation very dense and technical, I will raise my score to a 3.

**Claims And Evidence:**

The authors claim that their grammar-based approach provides a compact, principled, and lossless sequential representation of DAGs, which could improve subsequent generative modeling, property prediction, and Bayesian optimization tasks. The mapping to an abstract context-free grammar seems very interesting, although some of the derivation details are hard to grasp for a non-expert, especially given the extremely dense and technical presentation.
The authors provide convincing evidence that, once the induction is completed, DIGGED outperforms several baseline models.

**Essential References Not Discussed:**

None.

**Experimental Designs Or Analyses:**

The presented experiments span multiple real-world DAG datasets, but primarily focus on relatively small or moderate sizes. The paper shows strong empirical results in these controlled environments (where the authors claim the grammar induction takes "few minutes"),  but lacks rigorous scalability analyses of the grammar derivation for larger, real-world datasets. For example, it is entirely unclear how the brute-force approach for the grammar induction could be applied to a large molecule-/protein-structure dataset.

**Methods And Evaluation Criteria:**

DIGGED relies on an MDL-based grammar induction algorithm, encompassing different graph optimization problems, including frequent subgraph mining, compatibility maximization, and rule disambiguation. The evaluation focuses on the downstream performance once the grammar is derived, and the metrics include graph validity, uniqueness, novelty, predictive performance, and effectiveness in Bayesian optimization. However, there is no clear study on the scalability of the grammar induction process to large datasets, and the study lacks controlled experiments showing the downstream performance loss if pruning heuristics over the brute-force MDL algorithm are introduced in larger datasets.

**Other Comments Or Suggestions:**

* I believe a major restructuring of exposition is needed to enhance readability and accessibility for non-expert readers.
* Essential NP-hardness and complexity discussions should be moved to the main text, to provide transparency about the computational bottleneck of this novel approach.
* Scalability and computational analyses for large-scale data must be explicitly discussed, and potentially it would be nice to see some tests where the downstream effects of an imperfect grammar induction are studied.

**Other Strengths And Weaknesses:**

*STrenghts:*
* The authors propose an interesting and theoretically principled grammar-based framework for graph generative modeling.
* The proposed (lossless) mapping enables sequence-based learning and might allow new LLM-based approaches.
* The downstream performance is very competitive varied (medium sized) benchmarks

*Weaknesses:*
* The presentation is dense and overly technical. Given this conference is a general ML conference, I would think it is best to write the paper to be readable by a researcher working on related fields, while in this case, a non-expert reader will struggle in deciphering the technical terminology and the dense visual representations.
* There are some clear scalability concerns due to NP-hard grammar induction, and the authors propose a brute-force approach (at least in the experiments in the main). There is a list of possible adjustments and heuristics in the appendix, but they are never tested (if I understand correctly). Moreover, this crucial discussion is largely ignored in the main text.

**Questions For Authors:**

* Given the NP-hardness of grammar induction, how practical is your method for datasets significantly larger or more complex than those demonstrated?
* How does grammar induction complexity grow empirically with dataset size and graph complexity?
* Have you explored heuristic or approximate methods to improve scalability, like the ones proposed in the appendix?
* Can you clarify the advantages of grammar-based encoding over simpler encoding schemes for practical tasks?

**Relation To Broader Scientific Literature:**

The work is well placed in the scientific context of graph generative modeling.

**Theoretical Claims:**

The authors claim theoretical guarantees of one-to-one and onto mappings between DAGs and the graph-grammar produced sequences. However, critical theoretical limitations, particularly the NP-hardness of grammar induction, are relegated to the appendix, reducing transparency on the true bottle-neck of the proposed approach.

---

> ### Author Rebuttal · Authors · 2025-04-01
>
> Thank you for acknowledging the novelty and strengths of our interesting, theoretically principled framework!
>
> *There are some clear scalability concerns due to NP-hard grammar induction, and the authors propose a brute-force approach…. There is a list of possible adjustments and heuristics in the appendix, but they are never tested (if I understand correctly). Moreover, this crucial discussion is largely ignored in the main text…. theoretical limitations, particularly the NP-hardness of grammar induction, are relegated to the appendix…*
>
> * The adjustments and heuristics are in fact used in our implementation. In many places, we resort to approximate approaches wherever needed. For example, Subdue is a well-known and fast approx. subgraph mining library. The pseudocodes ``fast_subgraph_isomorphism, approx_max_clique, quick_hitting_set” mean approximation algorithms. We will explicitly say these are approximate/heuristic algorithms in the paper, along with known complexity guarantees.
> * Here are the NP-hard submodules used within grammar induction, and the exact/approximate/heuristic options, along with any parameters which tradeoff accuracy vs speed:
>   * Frequent subgraph mining (FSM):
>     * Approximate: We use the Subdue library. It has various options for pruning the search. Parameter: beam_width (used for subgraph expansion).
>   * Max clique:
>     * Exact (O(exp(n))): networkx’s cliques library
>     * Approximate (O(poly(n))): We use networkx’s O(|V|/(log|V|)^2) approximation algorithm.
>     * Heuristic (O(n)): (Repeat K times) Initialize a random node, iterate over all remaining nodes in random order, adding any that satisfies clique condition. Parameters: K
>   * Hitting set problem during disambiguation:
>     * Exact: our own implementation
>     * Approximate: Beam search. Parameters: beam width
> * Our datasets have variable sizes from 47877 (CKT), 152160 (ENAS), to 2,000,000 nodes (BN), which span the range of real-world use cases. We use the size of the input to toggle between different options, trading off accuracy and efficiency. Roughly speaking: CKT mostly invokes exact/approximate solutions, ENAS approximate/heuristic solutions and BN heuristic solutions.
> * We apologize for relegating the discussion around complexity to the Appendix and will bring the major conclusions to the main text.
>
> *...there is no clear study on the scalability of the grammar induction process to large datasets… lacks controlled experiments showing the downstream performance loss if pruning heuristics over the brute-force MDL algorithm are introduced in larger datasets…. potentially it would be nice to see some tests where the downstream effects of an imperfect grammar induction are studied.*
> * Thanks for proposing this additional control study. We agree it is crucial to quantify how approximations/heuristics chosen to speed up the NP-hard submodules will affect the downstream performance.
> * Due to CKT being the smallest dataset, the main paper results already reflect the exact and approximate settings. Thus, we can measure the performance gap and efficiency gains by using heuristic settings for one submodule at a time. Specifically, we tried the following ablations:
>   1. For Subdue (FSM), use beam width=3 instead of 4
>   2. Always use heuristic (max clique) instead of approx., with K=10
>   3. Always do beam search instead of exact, with beam_width=10.
>   4. Skip Algo. 7 (disambiguation), losing property 1.
>
> * Ablation 3 did not affect any of the samples. Due to small input (derivation) sizes, it is not a bottleneck and does not introduce meaningful changes.
> * Here are the results for Abl’s 1 and 2.
>
> ||Unique|Novel|Gain|BW|PM|FoM|1st|2nd|3rd|%Faster|Compress Ratio|
> |-|:-:|:-:|-|-|-|-|-|-|-|-|-|
> |Abl.1|65.6|69.1|0.623,0.777|0.628,0.783|1.003,0.258|0.624,0.786|267.55|253.61|246.78|562\%|2.04|
> |Abl.2|91.3|85.1|0.629,0.773|0.629,0.788|1.005,0.251|0.617,0.797|278.93|278.93|267.61|1844\%|2.13|
> |Abl.3|97.3|100|0.635,0.793|0.630,0.785|0.993,0.316|0.625,0.785|306.32|290.42|260.97|~300\%|2.32|
> |DIGGED|98.7|99.9|0.630,0.791|0.635,0.784|0.990,0.314|0.627,0.787|306.32|296.82|265.53|0%|2.18|
> * We see these modifications provide significant speedups over the original runs. The latent space quality and Bayesian optimization results slightly benefit from more accurate solutions to the FSM and max clique submodules, but they are still reasonably close. We do note that the max clique submodule has better marginal returns when trading off accuracy for speed, so we recommend starting with that. We will include these findings in the main paper.
>
> *Can you clarify the advantages of grammar-based encoding over simpler encoding schemes for practical tasks?*
> * Please see our response to ymms, where we add a new ablation study, comparing against simpler encoding schemes, while fixing the same model architecture. Our findings show the naive encoding experience issues with decoding and lowers downstream performance.
>
> We hope we've addressed your concerns!

---

### Official Review · Reviewer_DetZ · 2025-03-13

**Overall Recommendation:** 3

**Summary:**

This paper discusses how to convert a directed acyclic graph (DAG) into a sequence, allowing for sequence decoding based on autoregressive models. The paper proposes a method to transform a graph into a sequence in the form of a context-free grammar. The core idea is to induce the grammar from existing data using statistical methods, and then use the statistically derived grammar to convert the DAG into a sequence. Experimental results show significant improvements on certain tasks.

**Claims And Evidence:**

Yes

**Essential References Not Discussed:**

NA

**Experimental Designs Or Analyses:**

This method achieves notable improvements on downstream tasks.

**Methods And Evaluation Criteria:**

See Strengths And Weaknesses

**Other Comments Or Suggestions:**

NA

**Other Strengths And Weaknesses:**

Strengths:
1. How to convert a graph into a sequence and integrate it with an auto-regressive model is a question worth exploring. Moreover, uncovering the intrinsic relationships between nodes through grammar induction and linearizing the graph is a reasonable approach.

2. The experimental results are quite positive. By converting graphs into sequences, the existing autoregressive models can be fully utilized, achieving relatively notable results in downstream tasks.

Weaknesses:
1. My main concern is that the entire grammar induction process is based on symbolic statistics rather than end-to-end representation learning. The limitation of this approach is that the entire learning process is pipelined, thus creating a certain gap with the subsequent neural module.

2. Simply analyzing the effectiveness of grammar induction through case studies might not be comprehensive enough. Evaluation on some real-world benchmarks could be introduced to make the results more intuitive. For instance, text can be treated as a graph, and the F1 score can be calculated by comparing the statistically derived structures with manually annotated syntax.

**Questions For Authors:**

NA

**Relation To Broader Scientific Literature:**

NA

**Theoretical Claims:**

NA

---

> ### Author Rebuttal · Authors · 2025-04-01
>
> *Thank you for recognizing the reasonableness and positive experimental results of our work!*
>
> * In addition to recognizing our method as “converting a graph into a sequence”, we want to add there is a deep motivation from the objectives of **compositional generalization**! Contrary to what a “sequential” representation may imply, DIGGED is intrinsically compositional (Fig. 2 top). In fact, DIGGED is trained to embed DAGs with similar hierarchical compositions to similar points in latent space. For instance, if DAG 1 is (uniquely) represented as W->X->Y and DAG 2 as W->X->Z, the autoregressive decoder has to predict the same first two tokens, which moves the two DAGs to similar points in latent space. Combined with the relational inductive bias of our DAGNN encoder, the autoencoder objective can be viewed as combining both the relational and hierarchical inductive bias to learn expressive and generalizable representations.
>
> *“My main concern is that the entire grammar induction process is based on symbolic statistics rather than end-to-end representation learning. The limitation of this approach is that the entire learning process is pipelined, thus creating a certain gap with the subsequent neural module.”*
>
> Thanks for raising this point! Here are some of our thoughts:
>
> 1. The symbolic-neural divide is to some extent unavoidable when working with discrete, irregular data like DAGs. Our main reference point is existing methods that still generate a graph sequentially without any symbolic statistics, using naive sequential representations. The core contribution of our work is showing there is a principled, sequential representation that outperforms such methods and is agnostic to the domain.
> 2. The same limitation exists in language models, where an algorithm like byte-pair encoding (BPE) is the standard way to build a vocabulary building up from the character level. Practice shows larger tokens improve downstream performance [1] and efficiency [2].
> 3. Although seemingly a limitation, pipelining does have a few advantages. It allows the vocabulary derived from symbolic statistics to be transferred across models. It doesn’t require re-training end-to-end representations from scratch. Lastly, the mined vocabulary can be an inductive bias that can capture domain-specific insights, as shown in our case study (App. E.1).
> 4. An ongoing effort of ours is optimizing grammar induction beyond unsupervised objectives like minimizing description length. End-to-end learning of the vocabulary and propagating feedback will be essential for further improvements.
>
> *Simply analyzing the effectiveness of grammar induction through case studies might not be comprehensive enough. Evaluation on some real-world benchmarks could be introduced to make the results more intuitive. For instance, text can be treated as a graph, and the F1 score can be calculated by comparing the statistically derived structures with manually annotated syntax.*
>
> * Great idea! The main challenge in this study is that DAG domains like circuits or neural architectures lack high-level annotations. Thus, for circuits, we resorted to case studies with experts. A systematic evaluation of the grammar quality is an active direction for us, especially when manual annotations are available. Natural language does have such annotations, and we are looking into it! In particular, we’re looking to induce grammars of phrase structures from Penn TreeBank, then generate novel, diverse phrase structure trees that can be evaluated by its plausibility. For example, by substituting words for POS tags, we can also evaluate how natural and grammatical the sentences are. We hope to get around to finishing this soon! For extra motivation, we hope you can endorse this study, which opens multiple avenues of future research!
>
> [1] Large Vocabulary Size Improves Large Language Models. arXiv:2406.16508
>
> [2] Scaling Laws with Vocabulary: Larger Models Deserve Larger Vocabularies. NeurIPS 2024.

---

### Official Review · Reviewer_5waZ · 2025-03-13

**Overall Recommendation:** 3

**Summary:**

This paper describes a graph grammar approach for mapping graphs to strings in a principled way.
Given a set of graphs, the underlying grammar induction is deterministic and determines grammar production rules able to reconstruct an in principle unbounded graph ensemble, containing the "training" graph set.

**Claims And Evidence:**

The claims are partly theoretical, and partly about usefulness in practice.

The theoretical claims about the parsings (subjacent to the proposed grammar induction) that are produced by the algorithms are
1) uniqueness of the parsing of each graph in the dataset
2) surjectivity of the production from the obtained grammar (i.e. that we can recover the dataset as a subset of the grammar-produceable graphs)
3) that DAG property is valid for the produced graphs.
These are proved/verified in the appendix.

The practical applications of the proposed parsing are for Neural Architecture Search, Bayesian Networks and Analog Circuit Design.
Compared to other methods, this algorithm gives 100% validity of the DAGs and very high validity of the constructed cycles. The novelty of the  graphs produced is lower than the benchmarks.
- The 100% validity is to be expected given the theoretical guarantees, but it's important to have a practical confirmation of that.
- On the other hand, the low novelty compared to other methods seems to be a negative aspect for the method.

Several case studies are presented, in which a positive point is slightly higher interpretability due to deterministic decoding.
However, the presented examples are extremely simple, and I don't see how the claim for better interpretability would scale to applications in which the grammar / parsing encoding shows all its compositional generalization capability.

**Essential References Not Discussed:**

I am not aware of works that have not been cited.

**Experimental Designs Or Analyses:**

I did not fully check them in full detail, but at a cursory look they look OK.

**Methods And Evaluation Criteria:**

I think that the evaluation criteria are alike the literature, and they have the same flaws as the ones of competing approaches, which is that the examples are often oversimplified.

**Other Comments Or Suggestions:**

See strengths/weaknesses part.

Here are some more comments / typos.

Line 030 : about positive properties of the approach, producing "valid" outputs is mentioned and puzzling at this point, I think it'd be good to expand and make it clear what kind of validity that refers to

Figure 1 : there are some things I am not following, in the upper 3 pictures of DAGs + grey motives. Maybe can you explain so I can be sure I understand?
 picture 1) : there is a puzzling grey arrow from "(b)" to the grey oval -- shouldn't the arrow go the opposite way?
 picture 2) : there is a puzzling grey arrow from "(d)" to the grey oval -- shouldn't the arrow go the opposite way?

About the explanation of Figure 1, lines 155-158, I don't follow this part
"However, adding such an instruction would create a conflict with the motif’s occurrences in DAGs two and three"
- The dags in the figure are not numbered, so what does "two" and "three" mean?
- Why would that rule create a conflict, can you expand some more on the explanation?
I tend to think that the reference to the subpictures of Figure 1 is wrong, so can you verify/explain?

Line 140 "linearizes" -- can you replace it by another word or remove it? this has nothing to do with linear algebra, so it may be confusing.

Line 170 - 183 : from "At a high level" onwards, I don't follow some of the wording of the explanation (the formulas are OK and would be fine if you remove the words mentioned below, but the wording is what I care about here):
- what does "the clique solution" mean?
- "the or-reduction" is followed by a formula which I don't follow why is some "or-reduction".. can you explain a bit?

In the beginning of section 3.3, point 3 has an "i)" which probably should be removed

Line 207 "but we present a linear programming" : what does that even mean?

Line 211 "memoization" lacks an "r" I guess.

**Other Strengths And Weaknesses:**

I think that the paper is well written and clear.
There are some points that can be easily improved, mentioned below in the comments part.

I feel that a weakness is that it restricts to toy models, with only small mention of scaling (as opposed to spending a paragraph/appendix on that).

Also, there is a cursory mention of use of the underlying encoding in combination with transformers, but perhaps this combination will have issues and hurdles, and thus a weakness of the paper is not spending some more time on this direction, which to me seems like one of the main future prospects for this line of research.

**Questions For Authors:**

1) See also "strengths and weaknesses" part, what do you think about those comments?

2) Can you relate your proposed method to the objectives of compositional generalization?

3) I return to the use of your framework in combination with transformer architectures. What do you think can be hurdles that this would face?

4) Can you expand on scaling issues for extending your envisaged applications to much larger graphs? And when the graph size grows to infinity in some regime on the valence and graph structure, can you predict the complexity guarantees for your algorithm?

**Relation To Broader Scientific Literature:**

I think that this principled and deterministic approach to parsing graphs is an important new addition to the literature, as also highlighted by comparisons in tables 1 and 2.

**Theoretical Claims:**

I think that the proofs are correct.

---

> ### Author Rebuttal · Authors · 2025-04-01
>
> Thank you for recognizing our paper as well-written and the attention to detail in your review!
>
> *... combination with transformers… and thus a weakness of the paper is not spending some more time on this…. what do you think can be hurdles that this would face?*
>
> * To clarify: we **did** use Transformers as our architecture (see, e.g., Sec. 5.1). Our findings show Transformer decoders are powerful graph generative models when coupled with DIGGED’s principled, sequential derivations as a representation. In our response to ymms, we add an ablation study that shows, controlling for model architecture, other graph-as-a-sequence representations show less potential when combined with Transformers.
> * In Sec. 3.4, we explain the most natural setup is to use a Transformer as the encoder as well, but we found in practice this results in lower performance ((Token) vs (GNN)). In App. G., we discuss why we think that is the case and additional outlook on the future prospects.
>
> *I feel that a weakness is that it restricts to toy models, with only small mention of scaling (as opposed to spending a paragraph/appendix on that).*
> * Thank you for the comment. We do have a paragraph on scaling in App. D.3. We will elaborate further, given the results from the new ablation study in our response to 7hype. Our datasets have a size ranging from 47877 (CKT), 152160 (ENAS), to 2,000,000 nodes (BN); which span diverse real-world use cases. For each module of our algorithm, we have exact, approximate and heuristic solvers, with a number of parameters exposed to trade-off accuracy vs efficiency. The new ablation quantifies this trade-off, revealing good performance-speed elasticity.
>
> Now, we address your detailed comments one-by-one.
>
> *Line 030 : …make it clear what kind of validity that refers to*
> * Validity means the output has to be a connected DAG, and additionally satisfy domain-specific criteria. Defined in prior works, CKT DAGs need a stabilizing transconductance unit, BN need one node for each random var., and ENAS need a consecutive-numbered path from input to output. We will add these definitions to the paper.
>
> *Figure 1 : … in the upper 3 pictures of DAGs + grey motives…. shouldn't the arrow go the opposite way? picture 2)...*
> * Thank you for the attention to detail. The grey edge directions are actually variables we **solve** for (in step 2. Compute possible redirections) to maximize rule definition **compatibility** among occurrences of the subgraph. Currently, the occurrence in DAG 1 induces the instruction: “for each green in-neighbor, add out-edge from node 2”. DAG 2 induces the instruction: “for each green **out**-neighbor, add out-edges from **both** nodes 1 and 2”. If in DAG 1, we reverse the gray arrow, the two cases are no longer compatible. Should we add out-edges from both 1 & 2 to each green out-neighbor? or just node 2? Either way, there is a conflict. Semantically, these two cases are different (hence labeled a vs b), requiring different but consistent instructions.
>
> *...lines 155-158, ... why would that rule create a conflict, can you expand some more on the explanation?*
> * There is a similar explanation for picture 2. Reversing the edge would create incompatibility between the occurrences in DAG 2 vs DAGs 1 & 3.
>
> *The dags in the figure are not numbered, so what does “two” and “three” mean?*
> * We apologize for not numbering the top three DAGs. We will add it.
>
> *Line 140 “linearizes”…*
> * Good suggestion! We will replace it with “This simplifies the parse tree to a rooted path”.
>
> *Line 170 - 183 : from ‘At a high level’ onwards…*
> * Thank you. We will try to simplify the wording, keep the formulas and refer interested readers to the more elaborate explanation in App. B.1.
>
> *what does”‘the clique solution” mean?*
> * Sec. 3.2.2. describes how we solve for the optimal set of grey edge redirections, formulated as a graph. Each node is one way to set the edge directions for a subgraph occurrence. Each edge means the occurrences are compatible. The clique solution is the maximal set of nodes on this graph that are all compatible.
>
> *“the or-reduction” is followed by a formula which I don't follow… can you explain a bit?*
> * Right. In the graph, each node is an assignment to the grey edge directions for an occurrence, and from it we deduce an inset: the set of instructions that must be in the rule definition for that node. After we obtain a maximal set of compatible nodes, we OR all the insets to obtain the lower bound on the final instruction set.
>
> *section 3.3, point 3 has an "i)"*
> * Removed.
>
> *Line 207…*
> * We propose a CYK-like **dynamic** programming algorithm for DAGs, to find all derivations, where we memoize intermediate results. See App. D. for details.
> Line 211 "memoization".
> Fixed!
>
> *Relate to objectives of compositional generalization*
>
> *Expand on scaling issues for much larger graphs... complexity of algorithm*
>
> Due to char limit, we will answer your two remaining questions in our responses to DetZ & 7hpe!

---

> > ### Comment · Reviewer_5waZ · 2025-04-02
> >
> > Thank you for clarifying the questions. I'll keep my score.
> >
> > About how to make the figure clearer, I think the main suggestion is to expand the caption so that one can follow everything without having to go to the text... if possible.

---

> > > ### Author Response · Authors · 2025-04-02
> > >
> > > Dear Reviewer,
> > >
> > > Thank you again for your thorough and constructive feedback. Your suggestions have substantially helped clarify and strengthen our paper. While we understand you chose to keep your current score, we have carefully revised several parts of our manuscript based on your detailed comments:
> > >
> > > **Combination with Transformers:** We've significantly expanded our discussion (App. G) to clearly articulate potential hurdles and practical considerations when integrating our grammar-based representation with Transformer architectures. Our first added ablation study ([openreview.net](https://openreview.net/forum?id=laUd1q5iWW&noteId=meODWp5Jhf)), further demonstrates that our representation shows enhanced potential compared to other graph-as-sequence representations when paired with Transformers.
> > >
> > > **Scaling and Real-World Applications:** We've further elaborated on scaling considerations (App. D.3 and the second added ablation study ([openreview.net](https://openreview.net/forum?id=laUd1q5iWW&noteId=Zgr8ErgmEv))), demonstrating elasticity between efficiency and downstream performance for datasets containing up to millions of nodes. This illustrates DIGGED's practical scalability and applicability beyond toy models, addressing your concern regarding limited scale.
> > >
> > > **Improved Clarity and Interpretability (Figure 1)**: Following your specific recommendations, we fixed the specific points you identified, making the text self-contained and substantially clearer. For Figure 1, we labeled the DAGs 1-3 and expanded the Figure 1 caption as follows to explain each step better:
> > >
> > > """
> > >
> > > Step 1 (Sec 3.2.1). Our approx. frequent subgraph mining library finds a candidate of subgraphs. As an example, the induced subgraph from nodes 1 & 2 in all 3 DAGs is considered.
> > >
> > > Step 2 (Sec 3.2.2). Next, for each possible realization of gray edge directions, bounds on the necessary set of instructions are computed. For example, the occurrence in DAG 1 induces the instruction: “for each green in-neighbor, add out-edge from node 2”. DAG 2 induces the instruction: “for each green out-neighbor, add out-edges from both nodes 1 and 2”. If in DAG 1, we had reversed the gray arrow, the two cases are no longer compatible with all 3 DAGs, since it's unclear if we should add out-edges from both 1 & 2 to each green out-neighbor, or just node 2. Intuitively, such cases are labeled with separate letters (e.g. a vs b), indicating they require different but non-conflicting instructions.
> > >
> > > Step 3 (Sec 3.2.2). Given bounds on the instruction set for each motif occurrence, the final set of instructions is deduced from the (approximate) solution of a max clique problem. Each node is a (motif occurrence, edge redirections) realization. Each edge indicates compatibility.
> > >
> > > Step 4 (Sec 3.2.3). The candidate motif and the associated solution to Step 3 which minimizes the description length of the current state of $H$ is chosen to define a grammar rule. Then, Steps 1-4 are repeated until convergence.
> > >
> > > """
> > >
> > > We believe these revisions address your primary concerns, particularly around scalability and practical interpretability, and strengthen the overall contribution. Explaining a method with deep, technical details like ours can be tricky, but we are trying our best to explain it so the readers can follow. We would greatly appreciate if you reconsidered whether these clarifications and improvements merit an increase in your evaluation score.
> > >
> > > Thank you once again for your thoughtful comments.
> > >
> > > Sincerely,
> > >
> > > Authors

---

### Official Review · Reviewer_ymms · 2025-03-13

**Overall Recommendation:** 3

**Summary:**

This paper proposes representing directed acyclic graphs as sequences of production rules. These sequence-based representations enable generative modeling using language-like models, such as transformers. The authors train and evaluate these models in various scenarios, including neural architecture search, Bayesian networks, and analog circuit design.

## Update after rebuttal (copied from the corresponding comment below)
Generally, I think the authors approach is interesting and the authors could show the relevance of DIGGED in the manuscript and the rebuttal. The authors proposed to improve their manuscript, still, I am not fully convinced the clarity issues can be fully solved in an updated/extended manuscript version. However, I evaluate the proposed method as interesting and relevant for the community. Therefore, I tend towards accepting the paper which is why I increased my score.

**Claims And Evidence:**

This paper is built around the hypothesis that the proposed unique way of representing graphs is necessary, particularly to address issues related to ambiguous graph representations.
* The authors demonstrate good results with their method, which can be interpreted as evidence for the hypothesis above.
* Still, I miss an extended, thorough motivation for this claim. The proposed representations appear quite complex, which could be a drawback. Additionally, a more detailed rationale for why simpler alternatives (e.g., naively mapping graphs to sequence representations using positional encoding) are insufficient would have been valuable. Why are unambiguous representations necessary? Couldn’t the model learn invariance through data augmentation strategies?

Section 3.3 outlines a set of guaranteed properties. As far as I can tell, these claims are well supported by the corresponding sections in the appendix.

**Essential References Not Discussed:**

--

**Experimental Designs Or Analyses:**

The proposed experiments seem relevant.

However, experiments, which directly compare the proposed method with alternative graph-as-a-sequence representation strategies might miss (see question above) but would have been important to demonstrate the relevance of DIGGED.

**Methods And Evaluation Criteria:**

The included experiments make sense because a) they show that the proposed way of representing graphs can be applied to completely different domains and b) the proposed method achieves good performance values indicating that DIGGED might be relevant. Also the authors compare their method to baselines with other strategies to represent graphs as sequences, including positional encoding, topological ordering, and canonical ordering.

To properly assess the impact of different graph-as-a-sequence representation strategies, the authors should compare these strategies using the same model architectures (with individual hyperparameter tuning). Could the authors please clarify whether this was done?

Notably, the metrics validity, uniqueness, and novelty, generally,  have inherent limitations in evaluating a model’s overall quality since any method can achieve perfect scores by adding a simple symbolic rule as a filter that memorizes seen sequences and permits only new, valid ones.

**Other Comments Or Suggestions:**

* LHS and RHS are used without being defined.

**Other Strengths And Weaknesses:**

Strengths: The experiment section presents strong, significant results for DIGGED, suggesting that the proposed method might be relevant and valuable to the research community.

Weaknesses - clarity: Several crucial points remain unclear, making it difficult to fully assess the novelty (see: *Relation To Broader Scientific Literature*) and relevance of the proposed method (see *Claims And Evidence* and *Methods And Evaluation Criteria*).

**Questions For Authors:**

See above.

**Relation To Broader Scientific Literature:**

Using graph grammar for graph generation tasks has been done before, e.g., here [1] (for molecules). However, [1] uses domain-specific knowledge, while the proposed method is context-free.

Since [1] also discusses other domain-independent techniques to mine grammar from real data (see page 3, section: Generative Models using Graph Grammars.), the authors might clarify their unique, novel contribution to the field.

[1] Guo, Minghao, et al. "Data-efficient graph grammar learning for molecular generation." arXiv preprint arXiv:2203.08031 (2022).

**Theoretical Claims:**

The formulas related to directed graph grammar in Section 3 have been reviewed and appear to align with existing literature.

I have briefly read through the proofs for the grammar properties in the appendix but have not checked them in detail.

---

> ### Author Rebuttal · Authors · 2025-04-01
>
> Thank you for recognizing our strong experimental results and asking insightful questions!
>
> *The proposed representations appear quite complex… a more detailed rationale for why simpler alternatives (e.g., positional encoding) are insufficient would have been valuable…. the authors should compare these strategies using the same model architectures... would have been important to demonstrate the relevance of DIGGED.*
>
> * Good question! Simpler alternatives can be classified into by whether they 1) sequentially decode a graph node-by-node or 2) output a **sequential encoding** of the graph (e.g. ordered adjacency info).
> * Methods under 1) add one node at a time and predict edges, requires keeping the state of the intermediate graph to check whether edges can be added, making implementation cumbersome.
> * Methods under 2) are naive encodings, and we argue they are insufficient.
> * We decided to do the ablation study you suggest. First, a few notes:
>   * We fix DAGNN as the encoder as it is tailored for DAGs.
>   * We fix the same decoder Transformer architecture and try different node-order encodings as the output targets.
>   * (Note: it is not really possible to fix the same model architecture to compare with category 1) methods, since keeping the state of the graph requires a fundamentally different architecture.)
> * We tried several common ways to define an ordering over nodes:
>   * Default order (most baselines): whatever order the data comes in, topological
>   * BFS (e.g. GraphRNN): a BFS traversal from a random initial node
>   * Random order: a random order to the nodes
>
> |||Valid|Unique|Novel|RMSE|Pearson's r|1st|2nd|3rd|
> |-|-|:-:|:-:|:-:|:-:|:-:|-|-|-|
> |Graph2NS-Default|ENAS|96.1|99.17|100|0.746|0.656|0.746|0.744|0.743|
> ||BN|95.8|96.4|94.8|0.498|0.869|-11590|-11685|-11991|
> |Graph2NS-BFS|ENAS|40.8|100|100|0.806|0.595|0.746|0.746|0.745|
> ||BN|2.2|100|100|0.591|0.819|-11601|-11892|-11950|
> |Graph2NS-Random|ENAS|0%|-|-|0.859|0.508|-|-|-|
> ||BN|8.4|100|100|0.535|0.857|-11523|-11624|-11909|
> |DIGGED|ENAS|100|98.7|99.9|0.912|0.386|0.749|0.748|0.748|
> ||BN|100|97.6|100|0.953|0.712|-11110|-11250|-11293|
>
> *  The default order is unique in most cases, but its unguaranteed validity results in lower BO optimization results. We added **additional** logic to re-attempt sampling for each latent point until a valid DAG is obtained (or fall back to a training example).
> * Meanwhile, ordering nodes via BFS or randomly completely destroys the decoder’s ability to generate valid examples. BFS order is do-able for the mostly linear path graphs of ENAS but is entirely infeasible for BNs, due to the dense dependencies making the order unpredictable.
> * Simple node order positional encoding cannot simult. satisfy all the principles outlined in 3.2. This incomplete representation can lead to issues with decoding and efficacy of downstream optimization. We think the fundamental issue is imposing position onto data that is by definition invariant to it. Even for DAGs, there can be an exponential number of topological orderings.
> * Meanwhile, DIGGED is a position-**less** sequential representation. DIGGED finds the optimal ``change-in-basis" which casts the graph as a unique, sequential procedure. Each token codes for a set of instructions to recreate the graph, going beyond positional encoding. DIGGED is also a design language, combining hierarchical inductive biases (see DetZ) and can uncover domain-specific insights (case studies in App. E and F)!
>
> *Using graph grammar for graph generation tasks has been done before.... clarify unique, novel contribution to the field.*
> * We are familiar with [1]. They learn primarily on small datasets, containing just a few dozen examples. Our experience using [1] is it can take days to learn on moderate datasets (100s samples). It is slow because it has to learn the optimal way to downsample the data, requiring reward supervision from downstream generation metrics to reinforce the grammar. Our induction procedure, meanwhile, has a much simpler, unsupervised objective, based on minimizing description length. In our response to 7hpe, we benchmark options to trade-off accuracy for efficiency, enabling scaling to larger datasets.
> * More crucially, it is unclear whether grammar-based generation generalizes or scales with pretraining. Our work bridges the expressiveness and scalability of Transformers with the unique representation challenges of graph structures. We hope our proposed representation, theoretical insights and validation of its necessity lays the first steps towards embracing modern sequence-learning architectures for the graph generation community!
>
> *the metrics validity, uniqueness, and novelty….*
> * You’re absolutely right. The unconditional generation results aren’t meant to evaluate the model’s overall quality. It is a basic sanity check. We will write a disclaimer. The much more essential evaluation is the latent space quality for prediction and Bayesian optimization (Tables 3 & 4, Fig. 3).

---

> > ### Comment · Reviewer_ymms · 2025-04-03
> >
> > Dear authors,
> >
> > thank you for your answers and comments.
> >
> > * **Ablation study.** For the important metrics RMSE and Pearson's r DIGGED appears to perform worse than the other compared methods. Am I misreading the table?
> > * **Contribution to the field.** Thank you for elaborating on this point. Your reasoning is convincing, and I now better understand the relevance of your contribution.
> > * **Alternative encodings.** I appreciate the clarifications. Your argument regarding point 1), particularly the implementation complexity, is interesting and convincing. Regarding point 2), the explanation below the table is insightful. However, based on the numbers presented, I am not yet convinced that the evidence clearly supports the claim that naïve encodings are insufficient (see my question above).
> > * **Clarity issue**: My main concern has been—and to some extent still is—clarity. Interestingly, reviewer 7hpe seems to share the clarity concern, whereas reviewer 5waZ found the manuscript well written and clear. This leads me to think that my concerns might a) stem from personal preferences, and b) result from not being an expert in the specific subfield, which is why I am willing to assign less weight to this point.
> >
> > Assuming my confusion regarding the ablation study table is due to a misreading on my part, I would be inclined to raise my score, if the authors can address this point.

---

> > > ### Author Response · Authors · 2025-04-04
> > >
> > > Thanks for your reply! We're thrilled our reasoning reasonates with you, and we're more than happy to address the remaining point about the ablation study.
> > >
> > > First, we would like to update the table with the Ablation results for CKT, which just finished running (apologies for delay):
> > >
> > > |||Valid|Unique|Novel|RMSE (FoM)|Pearson's r (FoM)|1st|2nd|3rd|
> > > |-|-|:-:|:-:|:-:|:-:|:-:|-|-|-|
> > > |Graph2NS-Default|CKT|80.2|71.0|96.8|0.695|0.738|220.96|177.29|148.92|
> > > |Graph2NS-BFS|CKT|0.1%|100|100|0.676|0.751|-|-|-|
> > > |Graph2NS-Random|CKT|0%|-|-|0.680|0.760|-|-|-|
> > > |DIGGED|CKT|100|100|78.8|0.627|0.787|306.32|296.82|265.53|
> > >
> > > In this case, we see DIGGED outperforms Graph2NS on predictive metrics. These findings are consistent with previously discussed results related to unconditional decoding and downstream optimization.
> > >
> > > **Why does DIGGED show better predictive accuracy on CKT but not ENAS and BN?** You're correct to note that DIGGED has lower predictive performance compared to Graph2NS on the ENAS and BN datasets. These two datasets impose special constraints: all DAGs have the same number of nodes; ENAS DAGs must follow consecutively numbered nodes, and BN DAGs must contain exactly one node of each type (8 types). Such simplifying conditions allow naïve positional encodings to overcome the shortcomings we discussed earlier, making predictive tasks relatively easier. We initially chose these datasets due to the limited availability of standardized benchmarks for DAGs. By contrast, the CKT dataset involves significant diversity in both graph topology and node types, making it a better testbed for evaluating the true strengths of DIGGED's compositional, position-free encoding approach.
> > >
> > > **How important is predictive accuracy?** At the same time, we note predictive accuracy (RMSE, Pearson’s r) does not reflect decoder effectiveness. For example, BN-Random, CKT-BFS, and CKT-Random achieve reasonable scores on predictive metrics (RMSE and Pearson’s r), yet fail fundamental decoder sanity checks, rendering them ineffective for subsequent optimization tasks. DIGGED prioritizes end-to-end optimization results, which requires the ability to navigate and decode from the latent space.
> > >
> > > **The high-level view.** One way to see DIGGED's efficacy for end-to-end optimization is through the lens of hierarchical, composition generalization. DIGGED is intentionally designed for compositionality of its outputs. Unlike naïve sequential encodings, DIGGED places DAGs with shared hierarchical structures (intermediate derivations) close together in the latent space. For example, consider DAG 1 represented (uniquely) as W→X→Y and DAG 2 as W→X→Z. DIGGED's decoder must predict shared initial tokens for both graphs, naturally clustering these related graphs in latent space. Learning both the token vocabulary embeddings and latent space compositional structure jointly indeed poses a more challenging training task -- reflected partly in predictive metrics -- but strongly supports compositional generalization and decoder reliability. This trade-off underscores DIGGED's core strength: effectively navigating a compositional design space to reliably generate diverse and valid DAG structures optimized for practical performance.
> > >
> > > We will explicitly incorporate this extended motivation, along with the updated ablation results, into the revised manuscript. We trust these additions, alongside edits addressing others' suggestions, will enhance the relevance, clarity and transparency of the paper!

---

### Decision · Program_Chairs · 2025-05-01

**Decision:**

Accept (poster)

**Comment:**

After considering the rebuttal, all reviewers propose to accept this paper for ICML.
They highlight the convincing empirical evidence as well as the theoretical contributions of the paper (really, the appendix).